# Intratumor heterogeneity of EGFR expression mediates targeted therapy resistance and formation of drug tolerant microenvironment

Bassel Alsaed [1,2], Linh Lin [1,2], Jieun Son[3,4], Jiaqi Li[3,4], Johannes Smolander[1,2,5], Timothy Lopez[3,4], Pinar Ö. Eser[3,4], Atsuko Ogino [3,4], Chiara Ambrogio [6], Yoonji Eum[3,4], Tran Thai[3,4], Haiyun Wang [7], Eva Sutinen[8,9], Hilma Mutanen[1,2], Hanna Duàn [1,2,5], Nina Bobik[1,2], Kristian Borenius[10], William W. Feng [3,4], Behnam Nabet [11,12], Satu Mustjoki [1,2,5], Sanna Laaksonen[13], Benjamin K. Eschle[3,14], Michael J. Poitras[3,14], David Barbie[3,4], Ilkka Ilonen [2,10], Prafulla Gokhale [3,13], Pasi A. Jänne [3,4,15] & Heidi M. Haikala [1,2,3,4,15] ✉

Epidermal growth factor receptor (EGFR) tyrosine kinase inhibitors are commonly used to treat non-small cell lung cancers with EGFR mutations, but drug resistance often emerges. Intratumor heterogeneity is a known cause of targeted therapy resistance and is considered a major factor in treatment failure. This study identifies clones of EGFR-mutant non-small cell lung tumors expressing low levels of both wild-type and mutant EGFR protein. These EGFR-low cells are intrinsically more tolerant to EGFR inhibitors, more invasive, and exhibit an epithelial-to-mesenchymal-like phenotype compared to their EGFR-high counterparts. The EGFR-low cells secrete Transforming growth factor beta (TGFβ) family cytokines, leading to increased recruitment of cancer-associated fibroblasts and immune suppression, thus contributing to the drug-tolerant tumor microenvironment. Notably, pharmacological induction of EGFR using epigenetic inhibitors sensitizes the resistant cells to EGFR inhibition. These findings suggest that intrinsic drug resistance can be prevented or reversed using combination therapies.

Somatic mutations in *EGFR* occur in 10–50% of non-small cell lung cancers (NSCLC), varying from 10–15% in Caucasians to 30–50% in East Asian populations[1]. EGFR tyrosine kinase inhibitors (TKIs) have changed the treatment landscape for *EGFR* mutant NSCLC; however, most patients develop acquired drug resistance over time[2]. Osimertinib is a third-generation EGFR inhibitor binding more selectively to mutant EGFR, and it is now approved as the first-line treatment for metastatic and adjuvant NSCLCs with *EGFR* exon 19 deletions or exon 21 L858R mutations[3,4]. Although some of the genetic mechanisms of resistance to osimertinib are known, in 30–50% of the patients the mechanisms of

resistance are not understood[2]. Since drug resistance is typically studied after the cancer has relapsed, less is known about the cellular and microenvironmental events taking place during the formation of drug resistance, in the state called minimal residual disease (MRD) which forms following initial drug treatment. Instead of genetic resistance mutations that take longer to develop, the MRD state is most likely induced by intrinsic resistance mechanisms in the tumor cell population[5,6].

Intratumoral heterogeneity refers to genomic or biologic variation within a tumor lesion. This heterogeneity can exist either naturally or be

---

driven by evolutionary pressure, such as drug treatment, and it is a known enabler of therapy resistance[7]. Although some of the cellular heterogeneity arises from mutational mosaicism, cellular plasticity with dynamic changes in gene expression is considered as the functional contributor to tumor heterogeneity[8]. Intratumor heterogeneity of EGFR protein expression can be found from clinical samples of multiple cancer types[9–12]. Furthermore, reporter-based studies have revealed heterogeneity in the cell-to-cell EGFR activity in breast, colorectal and small cell lung cancer xenografts[13,14]. However, the biologic or therapeutic relevance of EGFR heterogeneity has not been addressed.

In this study, we identified pre-existing intratumoral heterogeneity of the EGFR expression within *EGFR* mutant NSCLCs, and that *EGFR*-mutant tumors harbor clones with low wild-type and mutant EGFR expression. We characterized the biology and therapeutic significance of these distinct EGFR expressing cell populations.

## Results

### EGFR protein expression is heterogeneous in *EGFR* mutant non-small cell lung cancers

While genetic heterogeneity has been acknowledged in *EGFR*-mutant NSCLCs, there is limited understanding of the heterogeneity at the protein level. In this study, we aimed to investigate the landscape of EGFR protein expression in tumors to assess the overall heterogeneity. We examined 8 patient-derived xenograft (PDX) samples originating from EGFR-mutant NSCLCs using immunohistochemistry (IHC) to evaluate EGFR protein expression in tumors (Fig. 1A, Supplementary Fig. 1A). We found that 50% (4/8) of the samples harbored distinct heterogeneous regions expressing low versus high levels of EGFR, while the remaining 50% (4/8) demonstrated diffuse staining patterns with no clear heterogeneity (Fig. 1A). Using PDX-expanded tumors, we assessed EGFR membrane expression in freshly isolated matched

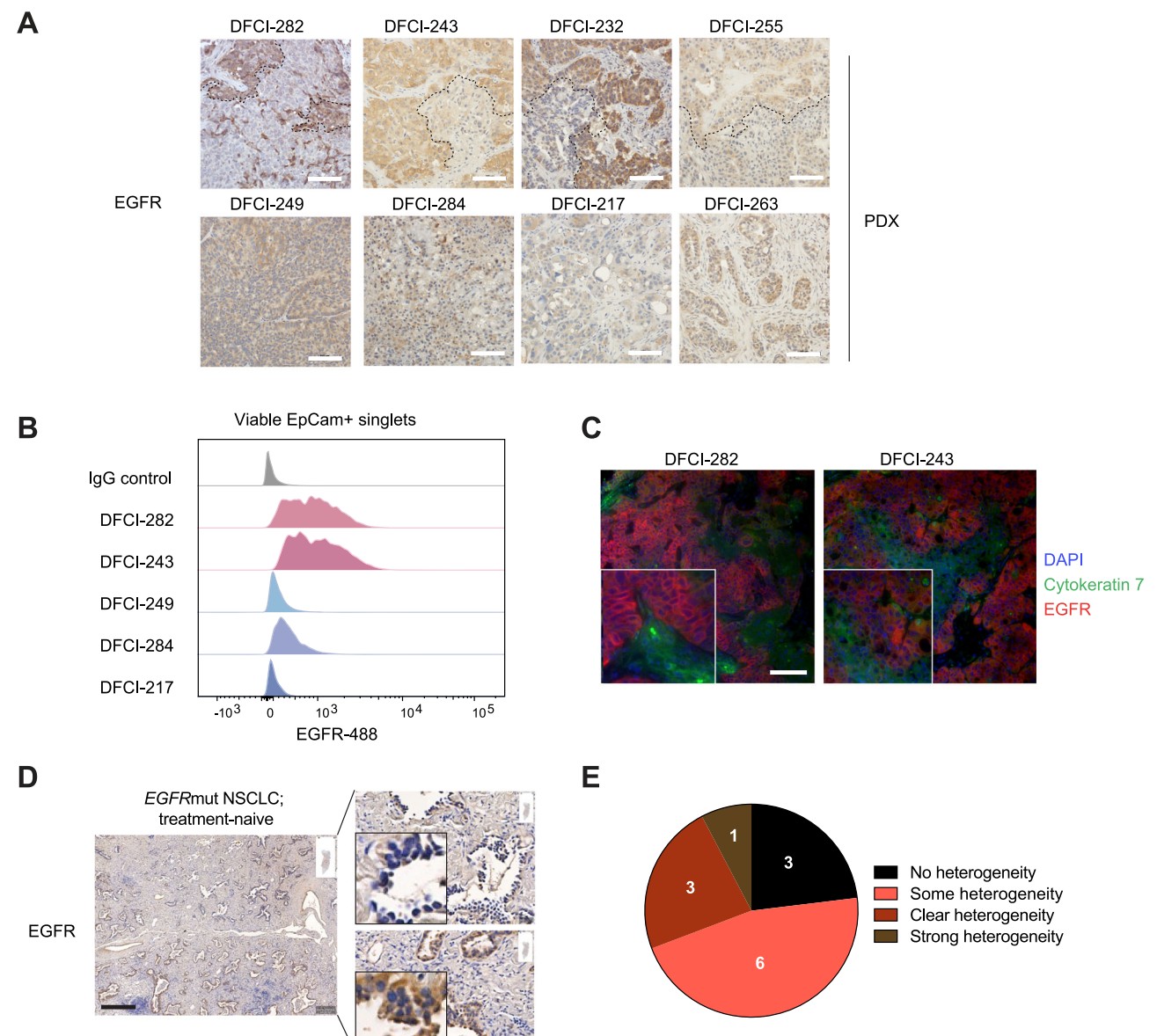

**Fig. 1 | Heterogenous expression of EGFR in *EGFR*-mutant NSCLC.**
**A** Immunohistochemistry staining for total EGFR in patient-derived xenograft (PDX) tumors shows heterogeneous expression of EGFR. The upper row displays PDX models with distinct EGFR heterogeneity, whereas the lower row shows samples with dispersed staining patterns. The black dotted line in the upper row highlights the areas of differential EGFR expression. Scale bar is 100 μm. **B** Flow staining for membrane EGFR in selected PDX models. The plot shows viable single epithelial (EpCam + ) cells. IgG-488 was used as a negative control. **C** EGFR-low cells are positive for the cytokeratin 7 (CK7) epithelial marker. DAPI marks the nuclei. Scale bar is 100 μm. **D** Treatment-naïve patients' tumors harbor heterogeneous regions for EGFR. Scale bar is 400 μm. **E** EGFR heterogeneity in 13 treatment-naive NSCLC patient samples. Heterogeneity was evaluated by a thoracic pathologist.

## Table 1 | Cell lines

| Cell line name | Source | Fingerprinting information |
|---|---|---|
| PC-9; human EGFR-mutant | Dr. Kazuto Nishio (Kindai University, Osaka, Japan) | Fingerprinted; RRID: CVCL_B260 |
| HCC4006; human EGFR-mutant NSCLC, male | ATTC | (CRL-2871); RRID: CVCL_1269 |
| H1975 | ATCC | (CRL-5908); RRID: CVCL_1511 |
| DFCI-284, DFCI-243, DFCI-169 | Established in the Jänne laboratory | N/A |

tumors by flow cytometry. The EGFR heterogeneity in viable epithelial cells (EpCam + ) was consistent with the dichotomous staining pattern observed in IHC, while less heterogeneity was present in the samples having diffuse EGFR staining patterns (Fig. 1B). To further confirm the epithelial identity of the EGFR-low expressing cells, we co-stained tumors with epithelial cytokeratin 7 and EGFR and found subpopulations of the tumor expressing the epithelial marker but having low expression of EGFR (Fig. 1C).

Since the stainings were done in PDX-tumors and some of the samples originated from heavily drug-treated patients, we wanted to know if the heterogeneous regions could be found directly from patient tumors prior to treatment. We retrieved tumor sections from 13 treatment-naive *EGFR*-mutant NSCLC patients and graded the level of EGFR heterogeneity (Fig. 1D, E, Supplementary Fig. 1B). EGFR heterogeneity by IHC was observed in the majority (77%) of the tissues originating directly from the patient tumors, with varying degrees of observable heterogeneity. The heterogeneity was not linked to a specific *EGFR* mutation, tumor type, or the tobacco smoking history of the patient (Supplementary Data 1). Expression heterogeneity was also observed in normal lung tissues and in tumors that were wild-type for EGFR, although the overall expression of EGFR in these tissues was relatively low. (Supplementary Fig. 1C–E). We conclude that *EGFR* mutant NSCLCs have heterogeneous expression patterns for the EGFR protein that can pre-exist in the tumors before the initiation of drug treatment.

### Isolation of the EGFR-low and EGFR-high expressing cells

To investigate if the observed heterogeneity in EGFR expression extends to cultured cell lines, we analyzed six cell lines derived from *EGFR*-mutant NSCLC patients, including three commercially available (PC-9, H1975, HCC4006) and three patient-tumor derived lines (DFCI-284, DFCI-243, DFCI-169) (Table 1). Cells were stained for membrane EGFR expression. We observed that while the baseline EGFR expression varied in each cell line, all cell lines had viable cells with very low EGFR expression (Supplementary Fig. 2A), indicating that the expression heterogeneity was recapitulated in the in vitro models.

To examine functional differences between EGFR-low and EGFR-high expressing cells, we sorted the cells based on their EGFR expression from three cell lines (PC-9, DFCI-284, H1975) using sequential cell sorting (Fig. 2A, C). We confirmed the identity of the cell lines by fingerprinting to exclude the possibility of cell line cross contamination. The established subpopulations had relatively stable EGFR protein status over passaging, although the expression slightly shifted over time (Supplementary Fig. 2B), and therefore the populations were maintained by sequential sorting. The subpopulations had differential expression of EGFR at both protein and mRNA levels (Fig. 2D–G, Supplementary Fig. 2C, D). We confirmed that the separation was not only for the total EGFR but also for the mutant EGFR using a mutant-specific EGFR exon19 E476-A750del antibody in PC-9 cells (Fig. 2H). We compared the allelic frequencies and *EGFR* copy number in PC-9 EGFR-low versus EGFR-high cells and found no

significant differences in either the allelic frequency of mutant/wild-type *EGFR* or the *EGFR* copy number (Fig. 2I, Supplementary Fig. 2E), suggesting that the difference in EGFR expression was caused by RNA or protein regulation rather than at the genomic alterations. EGFR gene signatures were enriched in EGFR-high cells compared to EGFR-low cells (Fig. 2J, Supplementary Fig. 2F), indicating that EGFR signaling was more abundant in EGFR-high cells.

### EGFR-low cells are intrinsically tolerant to EGFR inhibition

Next, we wanted to investigate how EGFR protein status influences the response to TKIs. In a short-term $IC_{50}$ assay, EGFR-low cells were slightly less sensitive to EGFR-inhibiting drugs osimertinib and pan-ErbB TKI afatinib, but there was no difference in sensitivity to pan-HER-targeting neratinib (Supplementary Fig. 3A, B), suggesting that the sensitivity difference was specific to EGFR inhibition. However, when exposed to osimertinib for a longer time, EGFR-low cells were significantly more resistant to the drug than EGFR-high cells, indicating that they were intrinsically more tolerant to the drug (Fig. 3A–C). The parental lines (with EGFR-high population as their majority clone) had similar sensitivity to osimertinib as the EGFR-high cells (Supplementary Fig. 3C). Regarding downstream signaling, EGFR-high cells had slightly more ERK phosphorylation, but this did not seem to impact on their sensitivity to MAPK inhibition (Supplementary Fig. 3D–F).

To address how EGFR expression relates to TKI sensitivity in general, we categorized 23 *EGFR* mutant lung cancer cell lines into EGFR-low and EGFR-high groups based on their *EGFR* mRNA expression and compared their $IC_{50}$ values with EGFR TKIs available in the CCLE database. We found that low EGFR expression was an overall predictor for poor EGFR TKI sensitivity with all the drugs tested (Fig. 3D, E, Supplementary Fig. 3G, Supplementary Data 2). This was also true when we limited the study to lung adenocarcinoma cell lines harboring only clinically acknowledged EGFR mutations (EGFR p.E756_A750del ELREA, L858R, T790M, Supplementary Fig. 3H).

To study the population interactions of the EGFR-low and EGFR-high cells under drug treatment, we labelled the cells with GFP and mCherry, respectively, and co-cultured them with or without constant exposure to osimertinib. As observed in the original cell lines, in an untreated 1:1 mixed co-culture EGFR-high cells became the major population over time, whereas the EGFR-low cells were maintained as a minor clone (Fig. 3F). However, when the cells were challenged with osimertinib, the EGFR-low cells were more tolerant to the drug and were able to outgrow the more drug sensitive EGFR-high cell population (Fig. 3G). Interestingly, although the EGFR-low cells were initially more tolerant to osimertinib, after multiple weeks of co-culture we observed the reappearance of the EGFR-high cells over time (Fig. 3H), which was not observed when the EGFR-high cells were grown as a single population.

To study long-term drug effects and to prevent wash-out of cells from the culture, we next placed the labelled cells in 3D tumoroid culture using different starting ratios. In both 1:1 seeded cells or when we mimicked the original PC-9 population with a 1:9 ratio (low:high), the EGFR-low cells were able to tolerate osimertinib better than the EGFR-high cells (Fig. 3I, J). EGFR-high cells were also more apoptotic than EGFR-low cells under osimertinib treatment in a dose-dependent manner (Supplementary Fig. 3I). As seen in the 2D culture (Fig. 3H), after weeks of culture with osimertinib the EGFR-high cells started to regrow (Fig. 3K), however, this took almost two months to occur. Also, the EGFR-high population regrew in the presence of EGFR-low cells, but not when the cells were cultured as population isolates (Supplementary Fig. 3J, K), suggesting that EGFR-low cells aid the formation of long-term drug resistance for both populations via an unknown mechanism.

Finally, we generated xenografts in mice using a 1:1 mixture of EGFR-low and EGFR-high PC-9 cells into immunodeficient mice. To study the MRD status, after tumor formation we treated the mice with

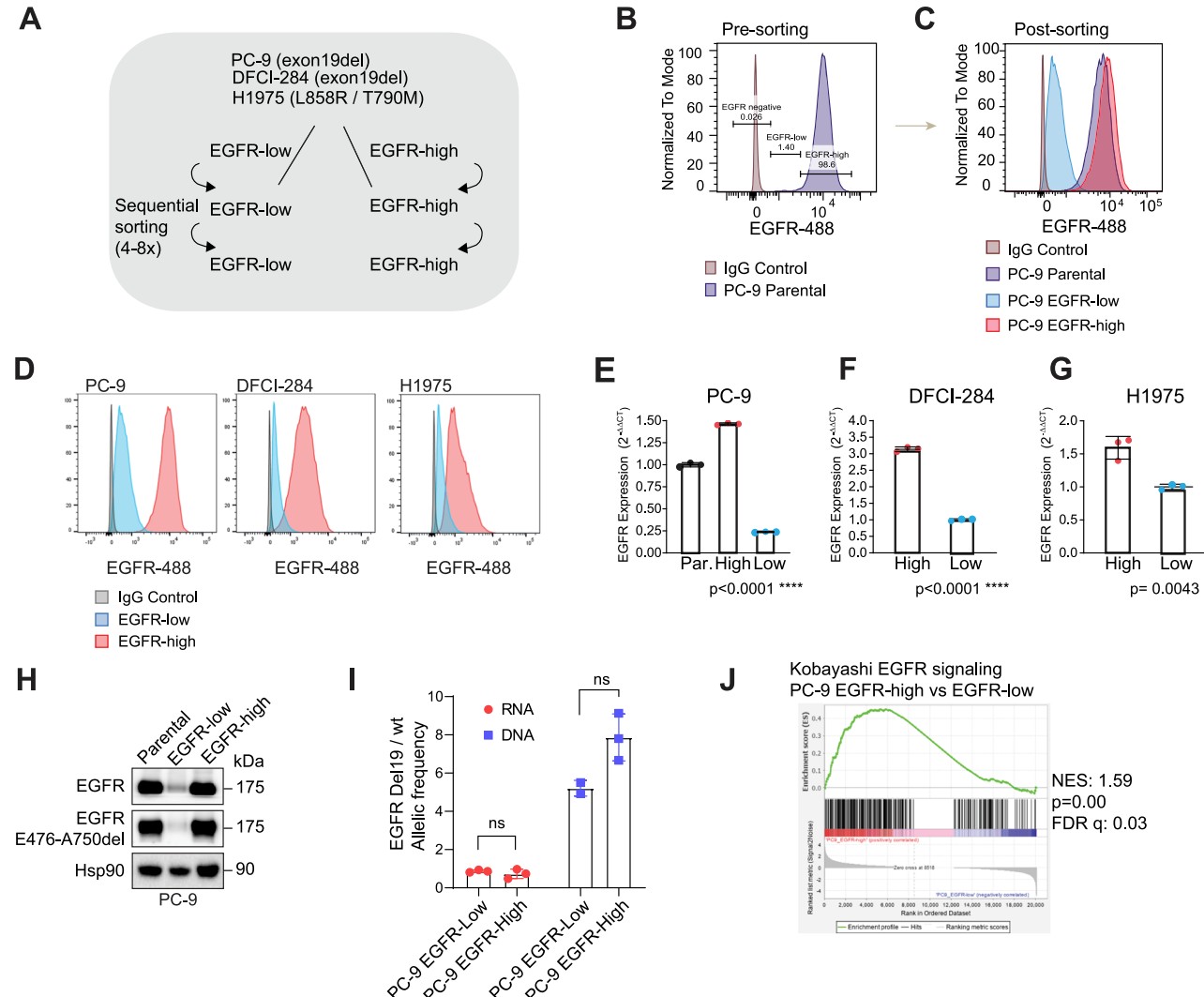

**Fig. 2 | *EGFR*-mutant NSCLC cell lines display heterogeneity in EGFR. A** Sorting scheme. Three *EGFR*-mutant cell lines were sequentially sorted for total EGFR using a viability marker. **B** EGFR expression in parental PC-9 cells before sequential sorting. IgG-488 was used as a negative control. **C** EGFR expression in PC-9 parental versus EGFR-low and EGFR-high cell lines after 6 rounds of sequential sorting. IgG-488 was used as a negative control. **D** EGFR expression in all three cell lines after 6 rounds of sorting. IgG-488 was used as a negative control. **E**–**G** *EGFR* mRNA expression in PC-9 / DFCI-284 / H1975, EGFR-high, and EGFR-low cell lines. *N* = 3 biologically independent experiments. **H** PC-9 EGFR-low and EGFR-high cells have differential expression on mutant EGFR. Western blot analysis showing total EGFR and EGFR E476-A750del expression. Hsp90 was used as a loading control. **I** PC-9 EGFR-low and EGFR-high cells have similar allelic frequencies. *EGFR*del19 / *EGFR*wt allelic frequency in PC-9 EGFR-low versus PC-9 EGFR-high cells. Allelic frequency was measured both in RNA and DNA level. *N* = 3 biologically independent experiments. **J** PC-9 EGFR-high cells are enriched with EGFR signaling compared to the EGFR-low cells. Gene-set enrichment analysis (GSEA). Three biological replicates. Data in (**E**, **F**, **G**, **I**) are presented as mean ± SD and analyzed by an unpaired student's *t*-test.

either vehicle or with two different doses of osimertinib (low-dose: 1 mg/kg, high-dose: 10 mg/kg). Tumors were collected for analysis on day 14 after the drug treatment (Supplementary Fig. 3L). While vehicle-treated tumors were dominated by EGFR-high cells resembling the events in the original cell line, the osimertinib-treated tumors consisted mainly of EGFR-low cells with only a few EGFR-high cells present in the tumors. Interestingly, the remaining EGFR-high cells had a stronger senescence-like morphology than what was observed in the EGFR-low cells, suggesting that they were possibly quiescent. In an additional experiment, the tumor-bearing mice were treated either with vehicle or with the high-dose osimertinib (10 mg/kg) for 21 days, and the population dynamics were followed over time by in vivo imaging (Fig. 3L). After the high-dose osimertinib treatment it took over a month for the tumors to re-grow, but again the EGFR-low cells were the ones to outgrow during the MRD (Fig. 3M, N).

In summary, our results demonstrate that EGFR-low cells are intrinsically more tolerant to osimertinib and thus more likely to

survive initial EGFR inhibitor treatment. However, the presence of EGFR-low cells also supports the maintenance and regrowth of EGFR-high cells during the formation of long-term drug resistance.

## EGFR-low cells upregulate integrin signaling to activate extracellular TGFβ

Since the co-cultures suggested that EGFR-high cells form drug resistant colonies more effectively in the presence of the EGFR-low cells, we further assessed the transcriptomic differences between EGFR-low and EGFR-high cells by RNA sequencing (PC-9 and DFCI-284). One of the top differentially expressed genes was *ITGB3* coding for Integrin beta chain β3, which was significantly upregulated in the EGFR-low cells than in the EGFR-high cells in both tested cell lines (Fig. 4A–C, Supplementary Fig. 4A). We also observed an upregulation in the mRNA levels of Integrin αV (*ITGAV*), which is a heterodimerization partner for Integrin β3 (Fig. 4D, E). When heterodimerized, integrins can activate growth factors from the ECM, and αVβ3 is known to activate latent

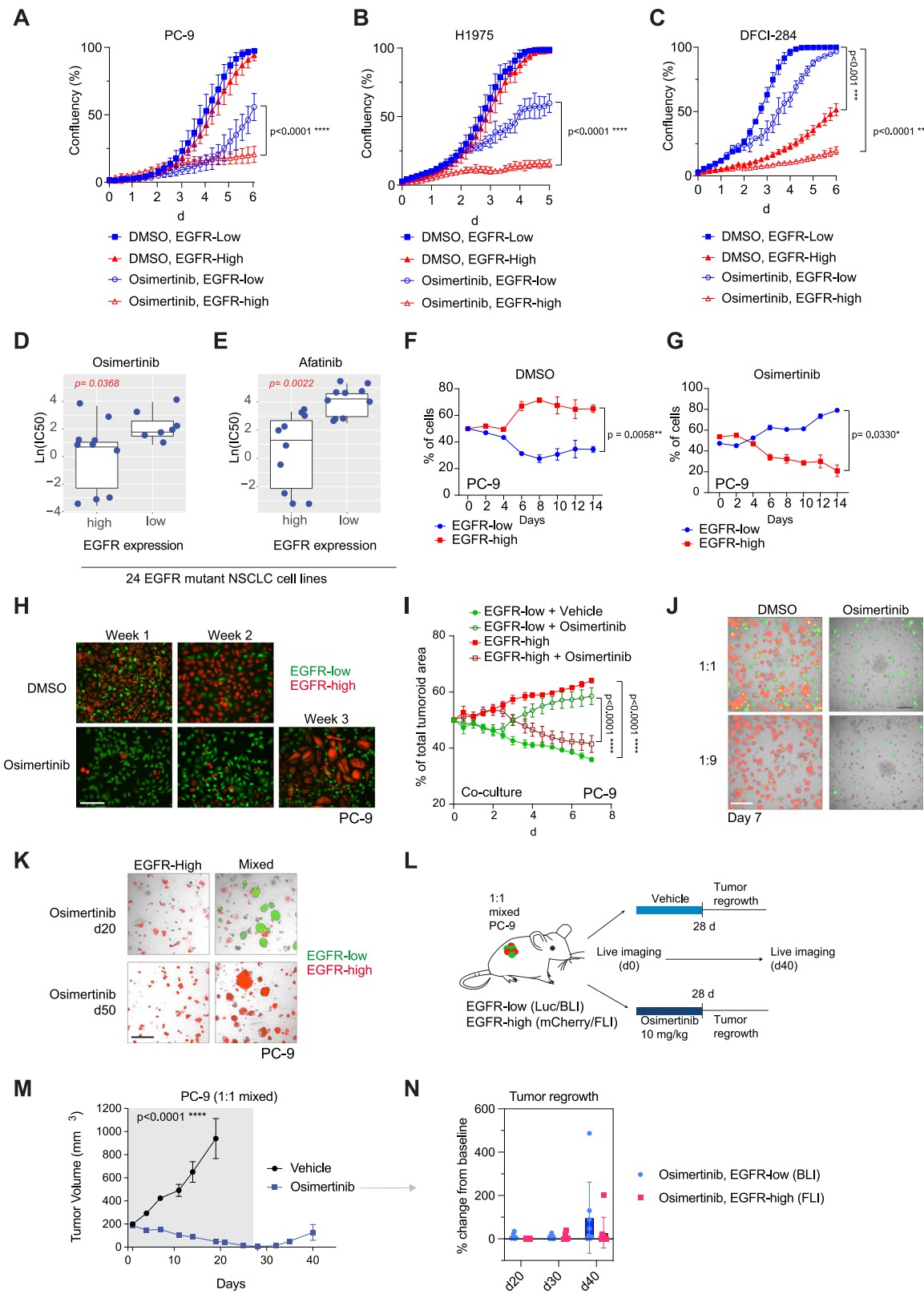

TGFβ (Fig. 4F)[15,16]. TGFβ has also been shown to induce positive feed-back regulation of αVβ3 and other integrins[17]. We compared the overall RNA expression levels of three different TGFβ isoforms *TGFβ1, TGFβ2*, and *TGFβ3* in EGFR-low and EGFR-high cells. There was no significant difference in the expression of *TGFβ1*, and *TGFβ3* was abundant in both cell lines. However, EGFR-low cells had significantly higher expression of *TGFβ2* in both tested cell lines (Fig. 4G, H). We also measured the

levels of activated TGFβ2 in the cell culture media and found a significant increase in the amount of activated TGFβ2 in the EGFR-low cell medium compared to the medium of EGFR-high cells (Fig. 4I), suggesting that EGFR-low cells secrete more TGFβ2 into their microenvironment. Furthermore, we observed an enrichment for TGFβ-regulated gene signatures, indicating increased TGFβ receptor signaling in the EGFR-low cells (Fig. 4J, Supplementary Fig. 4B, C).

**Fig. 3 | EGFR-low cells are tolerant to EGFR inhibition and enabling EGFR inhibitor resistance. A–C** EGFR-low cells are tolerant to osimertinib. $N = 3$ biologically independent experiments. **D** Low EGFR expression is associated with poor EGFR inhibitor response. Osimertinib sensitivity in cell lines categorized by high or low *EGFR* mRNA expression. Each dot represents an individual *EGFR*-mutant NSCLC cell line. **E** Afatinib sensitivity in cell lines categorized by high or low *EGFR* mRNA expression. Each dot represents an individual *EGFR*-mutant NSCLC cell line. **F**, **G** EGFR-low cells are tolerant to osimertinib over time. Labelled EGFR-low and EGFR-high cells were mixed in 1:1 ratio, and the population ratios were followed-up over time using flow cytometry. $N = 3$ biologically independent experiments. **(H)** EGFR-low cells are intrinsically tolerant to osimertinib, but EGFR-high cells can regrow over time in mixed culture. 10 nM osimertinib was used. Scale bar is 1000 µm. **I** EGFR-low tumoroids are more tolerant to osimertinib over time. 1:1 mixture of EGFR-low:high cells was used and the cells were treated with vehicle or 10 nM osimertinib. $N = 3$ biologically independent experiments each consisting of three technical replicates. **J** EGFR-low cells are the initial cell type of resistance in 1:1 (low:high) and 1:9 (low:high) mixed cells. Cells were treated with vehicle DMSO or 10 nM Osimertinib in tumoroid culture. Scale bar is 400 µm. **K** EGFR-high cells can form drug resistance grow over time, but only in mixed cultured. Cells were treated with 10 nM osimertinib and followed over time. Scale bar is 250 µm. **L** Treatment scheme for PC-9 EGFR-low:high xenograft. 1:1 ratio of cells was grafted into mice, and after tumor formation the mice were treated with either vehicle or osimertinib for 14 days, after which the tumors were collected for analysis. **(M, N)** EGFR-low cells escape drug treatment earlier than EGFR-high cells. In vivo measurement of labelled EGFR-high (mCherry) and EGFR-low (Luciferase) cells in xenograft mice. Data dots represent individual mice, $N = 8$ mice per group. Data in (**A, B, C, F, G, N**) are presented as mean ± SD and as mean ± SEM in (**M, I**). Data were analyzed by two-way ANOVA, Tukey's test in (**A, B, C**), by a paired student's *t* test in (**F, G**), and by tow-way RM ANOVA in (**I, M**). $N = 24$ in (**D, E**).

TGFβ is a well-known inducer of epithelial-to-mesenchymal (EMT) transition, so we investigated whether EGFR-low cells would exhibit more EMT-like phenotype. Indeed, EGFR-low cells had lower levels of epithelial E-cadherin and increased expression of smooth muscle actin (SMA), suggesting a more EMT-like phenotype (Fig. 4K). Additionally, EGFR-low cells exhibited reduced expression of proapoptotic BIM, potentially explaining their reduced apoptotic sensitivity to osimertinib. TGFβ is secreted by the cells in a latent complex consisting of TGFβ, latency-associated peptide (LAP) and latent-transforming growth factor beta-binding protein (LTBP). LTBPs interact with fibrillin and fibronectin in the ECM, and excess TGFβ can be stored in the ECM in its latent form[17]. Fibronectin, encoded by the *FN1* gene, is also a known mesenchymal marker and regulates cell survival and motility mainly through integrin signaling[18]. EGFR-low cells presented a significant upregulation of *FN1* (Fig. 4L), and fibronectin fibers were more visible in tumors originating from mice xenografted with only EGFR-low cells compared to mice xenografted with EGFR-high cells (Fig. 4M, Supplementary Fig. 4D), suggesting that the presence of EGFR-low cells can induce fibronectin production and fibrotic microenvironment. Gene signatures related to fibronectin matrix formation and ECM receptor interaction were also enriched in the EGFR-low cells (Supplementary Fig. 4E). Interestingly, the LTBP1 scaffold necessary for TGFβ secretion and storage in the ECM was also highly expressed in EGFR-low cells, and generally high *LTBP1* expression was associated with poor EGFR TKI response (Supplementary Fig. 4F, G). We also observed, that the genetic knock-out of either *ITGB3*, *FN1*, or *TGFβ2* was sufficient to make the EGFR-low cells more sensitive to osimertinib (Fig. 4N). Expression of EGFR was also increased in EGFR-low cells upon knockout or knock down of the three genes, especially with the knock down of *TGFβ2* (Supplementary Fig. 4H, I).

Overall, the data suggests that TGFβ2 secretion and ECM modulation induced by the EGFR-low cells can contribute to reduced EGFR inhibitor sensitivity.

## EGFR-low cells are invasive and modulate the tumor microenvironment

TGFβ is a pleiotropic modulator of the tumor microenvironment that is associated with increased cell migration and invasion, so we next assessed the migration properties of the cells by using trans-well migration assays. We detected significantly more EGFR-low cells migrating through the trans-well membrane in both tested cell lines, indicating higher motility of the EGFR-low cells (Fig. 5A–C). To further evaluate the invasion properties of the two populations, we used a microfluidic device platform[19] where collagen was loaded to the middle channel, chemoattractant serum was loaded into the distant side channel, and serum-deprived EGFR-low or EGFR-high cells were loaded to the proximal side channel (Fig. 5D). Cell invasion from the proximal side channel to the collagen matrix was measured over time. We noted that EGFR-low cells were more capable in invading the ECM than EGFR-

high cells, suggesting higher invasive potential (Fig. 5D, E). Interestingly, knocking down *ITGB3*, *FN1*, or *TGFB2* each effectively reversed the invasion observed in the microfluidic channels. The most pronounced change in phenotype was observed with the knockdown of *TGFB2*, followed by *ITGB3* (Supplementary Fig. 5A, B).

Interestingly, when we analyzed the tumors from mice grafted with a 1:1 mixture of the labelled EGFR-low/high cells, the EGFR-low cells were residing in the edges of the tumor, suggesting that they might be responsible for forming the invasive or protective front on the edge of the tumor (Fig. 5F). When comparing the amounts of distal metastasis detected in the histologically stained mouse tissues, there was more metastasis in the organs of the EGFR-low grafted mice compared to EGFR-high grafted mice, demonstrating the metastatic potential of the EGFR-low cells in vivo (Supplementary Fig. 5C, D).

Since TGFβ also modulates the tumor microenvironment, we next tested the microenvironmental effects of EGFR-low versus EGFR-high cells. Tumors arising from EGFR-low cells had a stronger fibrotic phenotype than EGFR-high tumors, and the fibers were highly positive for SMA (Supplementary Fig. 5E), suggesting that the EGFR-low tumors were more EMT-like, but also potentially harboring more cancer-associated fibroblasts (CAFs). To study how EGFR-low and EGFR-high cells directly interact with fibroblasts, we isolated CAFs from a patient tumor and studied the cell-cell interactions utilizing the microfluidic device (Fig. 5G, H). This time, the tumor cells were seeded in the middle chamber of the microfluidic device, whereas a fibroblast-PBS mixture was loaded into the side channel. 20% serum was added to the distal side channel to increase chemo-attraction to the middle channel. Fibroblast migration from the side channels towards the cells was measured over time. Interestingly, there was significantly higher attraction of fibroblasts to the middle chamber induced by the EGFR-low cells (Fig. 5H-JI), suggesting that EGFR-low cells can also modulate their TME by fibroblast recruitment. Moreover, ectopic addition of TGFβ2 was able to enhance cell proliferation (Supplementary Fig. 5F).

Finally, since TGFβ is also known for its immunosuppressive functions, we wanted to address if the presence of EGFR-low cells affects antitumor immunity. Importantly, when we analyzed the immune-related cytokines from the medium secreted by EGFR-low versus EGFR-high cells, we noticed that multiple immunosuppressive cytokines were secreted more by the EGFR-low cells, including CCL-2, CCL-5, IL-4 and IL-8 (Supplementary Fig. 5G). To functionally test whether EGFR-low cells would extract more immunosuppressive behavior, we placed the cells under natural killer (NK) cell treatment and tested the apoptotic effect of NK cells after 6 and 24 h. The EGFR-low cells were able to resist the NK-cell-dependent killing better than the parental or EGFR-high cells (Fig. 5J, K), suggesting that they have higher capacity to resist immune cell killing.

As a summary, we hypothesize that EGFR-low cells are more prone to survive under EGFR-TKI treatment and thus more dominantly present during the MRD state. The surviving EGFR-low cells attract

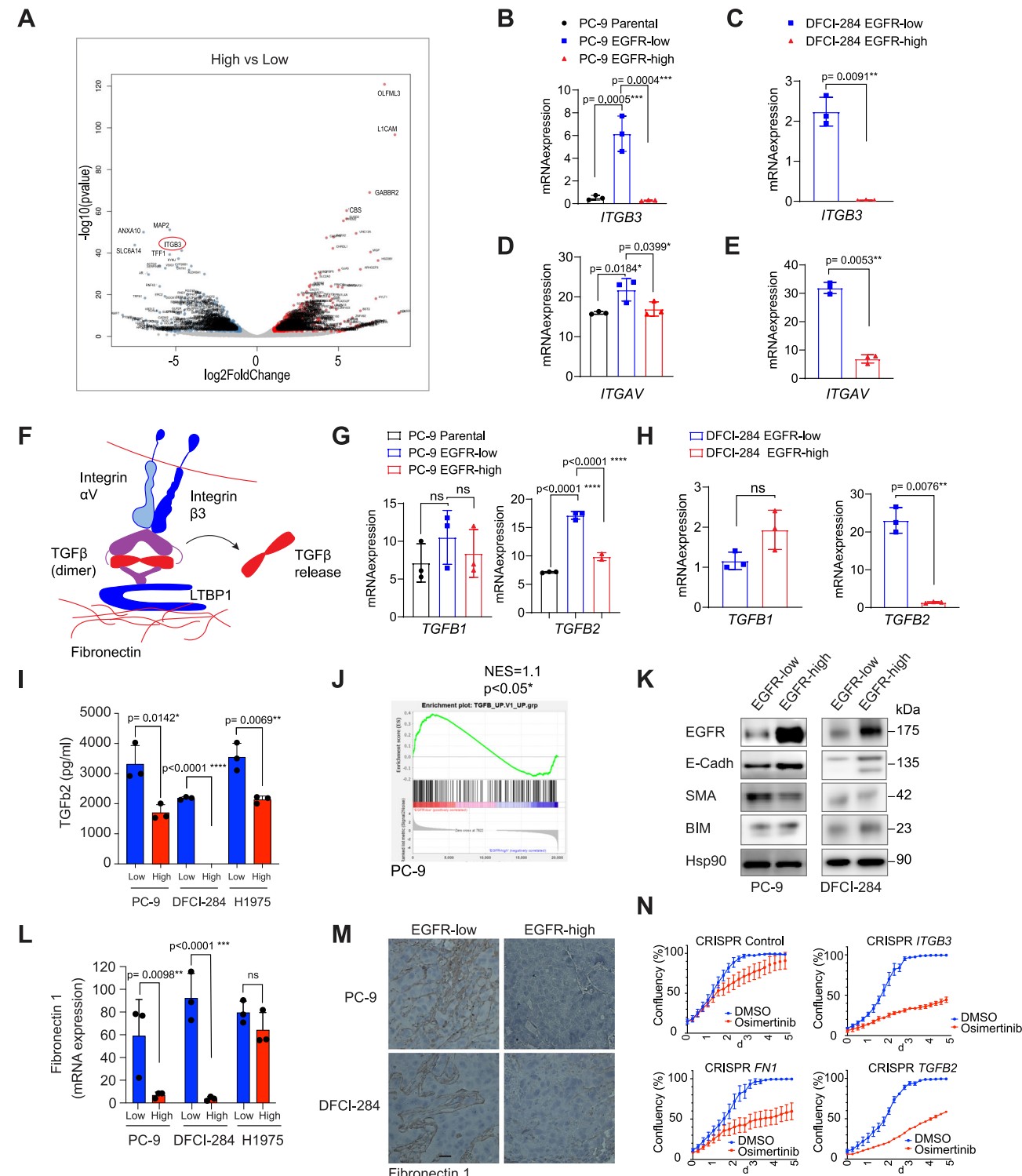

fibroblasts, secrete immunosuppressive cytokines, and actively contribute to the formation of the drug resistant niche, finally enabling the regrowth of the tumor cells (Fig. 5L).

**Epigenetic inhibitors alter the cellular state of the EGFR-low cells**
Since the EGFR expression between the cell lines was not markedly different at the level of genetic alterations, we hypothesized that EGFR was regulated at the epigenetic level, and that the low EGFR expression status could potentially be reversed by epigenetic modifiers. We screened nine epigenetic inhibitors with various mechanisms of action and measured their $IC_{50}$ values and effects on EGFR membrane

expression. We noted slightly increased EGFR membrane expression with EZH-2 inhibitor (EPZ-6438), but EGFR upregulation was more striking using a pan-HDAC inhibitor (SAHA/vorinostat) (Fig. 6A). The HDAC inhibitor induced EGFR expression was prevalent in all tested cell lines (Fig. 6B–D). HDAC inhibition also reversed the EMT-like phenotype indicated by reduced SMA and increased BIM expression (Fig. 6B). EGFR-low cells were selectively more sensitive to HDAC inhibition than EGFR-high cells (Fig. 6E, F, Supplementary Fig. 6A). Additionally, when PC-9 cells were blotted with the EGFR mutation specific antibody, we noted that vorinostat was increasing the expression of both total and mutant EGFR (Supplementary Fig. 6B, C).

**Fig. 4 | EGFR-low cells exhibit increased TGFβ expression and secretion, leading to a more mesenchymal phenotype. A** Volcano plot comparing RNA sequencing data from PC-9 EGFR-high versus EGFR-low cells. The red circle highlights *ITGB3*. $N = 3$ biologically independent experiments. (**B–** EGFR-low cells express higher levels of *ITGB3* and *ITGAV* mRNA in both PC-9 EGFR-low and DFCI-284 EGFR-low cells compared to the EGFR-high cells. $N = 3$ biologically independent experiments. **F** Schematic illustrating the activation mechanism of latent TGFβ. TGFβ is secreted by cells in a complex with LTBP, and excess TGFβ can be stored in the ECM. Heterodimerized αVβ3 integrins activate TGFβ from the ECM by inducing mechanical forces, after which the released TGFβ binds its receptors to activate downstream signaling. **G, H** RNA expression of *TGFB1* and *TGFB2* in PC-9/DFCI-284 EGFR-low and EGFR-high cells. *TGFB3* was undetectable. $N = 3$ biologically independent experiments. **I** EGFR-low cells secrete more activated TGFβ2 than EGFR-high cells. Secretion of activated TGFβ2 was measured using ELISA. Medium was collected after 3 days of culture. Low: EGFR-low cell lines, High: EGFR-high cell lines. $N = 3$ biologically independent experiments. **J** Enrichment of TGFβ-signaling in PC-9 EGFR-low cells. GSEA with 3 biologically independent experiments. **K** EGFR-low cells exhibit a more EMT-like phenotype. Hsp90 was used as a loading control. **L** EGFR-low cells express more *FN1* mRNA in all three cell lines. $N = 3$ biologically independent experiments. **M** PC-9 EGFR-low xenograft tumors are more positive for fibronectin 1 than EGFR-high tumors. Scale bar is 200 μm. **N** CRISPR-based knockout of *ITGB3*, *FN1*, or *TGFB2* sensitizes EGFR-low cells to osimertinib. Cells were treated with vehicle DMSO or 10 nM osimertinib. $N = 2$ biologically independent experiments each consisting of four technical replicates. Data in (**B, C, D, E, G, H, N**) are presented as mean ± SD and as mean ± SEM in (**I, L**). Data were analyzed by one-way ANOVA, Dunnett's test in (**B, D, G**), by a paired student's *t*-test in (**C, E, H**), an unpaired student's *t*-test in (**I**), and by two-way ANOVA in (**L**).

E-cadherin expression was not altered, suggesting only a partial phenotypic restoration by the treatment (Supplementary Fig. 6D).

Since vorinostat is a pan-HDAC inhibitor targeting all class I and class II HDACs[20], we compared the expression of different HDACs in EGFR-low versus EGFR-high cells, and found higher expression of HDACs 1-3 in both tested EGFR-low cell lines in both RNA (Fig. 6G, H) and protein level (Fig. 6I, J). There was overall higher histone deacetylase activity in the EGFR-low cells portrayed by the absence of histone 3 acetylation (H3-Ac). Knock-down of HDACs 1-4 by siRNA were individually inducing higher expression of EGFR (Fig. 6K), but the effect was most prevalent with siHDAC1. Lastly, the TGFβ gene signatures were abolished by vorinostat treatment (Fig. 6LN), and TGFβ2 secretion to the medium was hindered (Fig. 6O). Furthermore, vorinostat was able to inhibit the attraction of CAFs by EGFR-low cells in the microfluidic device (Fig. 6P), suggesting that the drug was beneficial also for reversing the microenvironmental effects caused by the EGFR-low cells.

## Pharmacologically induced EGFR sensitizes cells to EGFR inhibition and prevents drug resistance

We investigated whether the phenotype change induced by HDAC inhibition could alter the sensitivity of EGFR-low cells to EGFR inhibition. As we had previously observed, single-agent osimertinib was ineffective against EGFR-low cells, while EGFR-high cells were sensitive to the drug (Fig. 7A, B, Supplementary Fig. 7A, B). However, when EGFR-low cells were co-treated with vorinostat, the combination delayed the regrowth of the cells, and reversed the impaired apoptosis in the cells (Supplementary Fig. 7C, D). Drug synergy assays confirmed a clear synergistic effect of osimertinib with vorinostat, with EGFR-low cells being especially sensitive to the combination of drugs (Supplementary Fig. 7E–G). However, better synergy with a lower drug dose was obtained with panobinostat, another HDAC inhibitor which has been clinically approved by the FDA for the treatment of multiple myeloma (Fig. 7C). In a long-term follow-up, osimertinib + panobinostat prevented the regrowth of the cells with a lower drug dose compared to osimertinib + vorinostat, indicating that panobinostat was more potent (Fig. 7D–F). Based on these findings, we chose to use panobinostat for the in vivo studies. Mechanistically, although the effect of HDAC inhibitors on EGFR expression is indirect, we observed that ectopic expression of EGFR alone was sufficient to sensitize EGFR-low cells to osimertinib, while degradation of the ectopic EGFR using dTAG system restored osimertinib tolerance (Supplementary Fig. 7H–J).

To address the in vivo efficacy of the EGFR phenotype conversion, we generated 1:1 mixed EGFR-low and EGFR-high cells xenografts and the tumor-bearing mice were pre-treated for 24 h with either vehicle or panobinostat (5 mg/kg). The mice were then continuously treated with either vehicle, osimertinib (10 mg/kg), panobinostat (5 mg/kg) or osimertinib + panobinostat for 28 days (Fig. 7G). The mice were followed up until day 60. As previously seen in vitro, administration of panobinostat was increasing the EGFR expression also in vivo (Fig. 7H). During the treatment period we observed tumor growth suppression with both single agent osimertinib and with the combination of osimertinib+panobinostat. However, after the drug withdrawal we started to observe differences in tumor regrowth (Fig. 7I). During the follow-up period, the combination of osimertinib and panobinostat associated with the lowest number of mice with tumor regrowth (Fig. 7I), and mice treated with single agent osimertinib had more recurring tumors than combination treated mice (Fig. 7J). Single agent panobinostat had almost no effect to the tumor growth.

Finally, to further investigate the therapeutic responses of EGFR-mutant cancers, we utilized a novel tumor-on-a-chip (TOC) system. This system incorporated two distinct lines of patient-derived tumor organoids (PDTOs) harboring EGFR mutations. Our TOC system is designed to simulate critical aspects of the in vivo tumor microenvironment, including drug permeability and interactions with tumor cells growing in three-dimensional extracellular matrix (3D-ECM). A crucial element of this model is the endothelial barrier integrated within the chip (Supplementary Fig. 7F), which therapeutic agents must traverse to reach the tumor organoids. This feature not only complicates the drug delivery process but also enhances the model's biological relevance by closely mimicking physiological conditions.

Within the TOC, the EGFR-mutant PDTOs demonstrated increased responsiveness when treated with a combination of osimertinib and panobinostat compared to single agents (Fig. 7K). The results indicate that combining osimertinib with panobinostat provides significant therapeutic benefits in a model that closely represents patient-derived conditions.

Our findings suggest potential clinical applications for enhancing treatment efficacy in EGFR-mutant cancers. Despite the enhanced treatment outcomes, in mice, some tumors regrew regardless of the combination treatment, indicating that other TKI escape mechanisms might have taken place in the persisting tumors. We conclude that minimizing the observed EGFR heterogeneity by epigenetic inhibition could be a viable therapeutic strategy to prevent EGFR TKI tolerance and formation of drug resistant microenvironment during MRD state (Fig. 7J).

## Discussion

Cell-to-cell heterogeneity in genetic alterations ("mutation mosaicism") of oncogenic *EGFR* has been suggested to impact EGFR inhibitor sensitivity[21,22] and different *EGFR* mutations have varying sensitivities to TKIs[23,24]. It has been observed that even minor tumor subclones can substantially contribute to treatment outcomes[25]. While some heterogeneity can arise from having multiple mutations in the tumor, *EGFR* driver mutations are considered as an early event in tumor evolution[26,27], and mutational mosaicism is rarely detected in patient tissues[28–30]. While the focus of tumor heterogeneity studies has recently shifted from genetic mutations towards addressing cellular plasticity and heterogeneity in protein expression[31], our study suggests

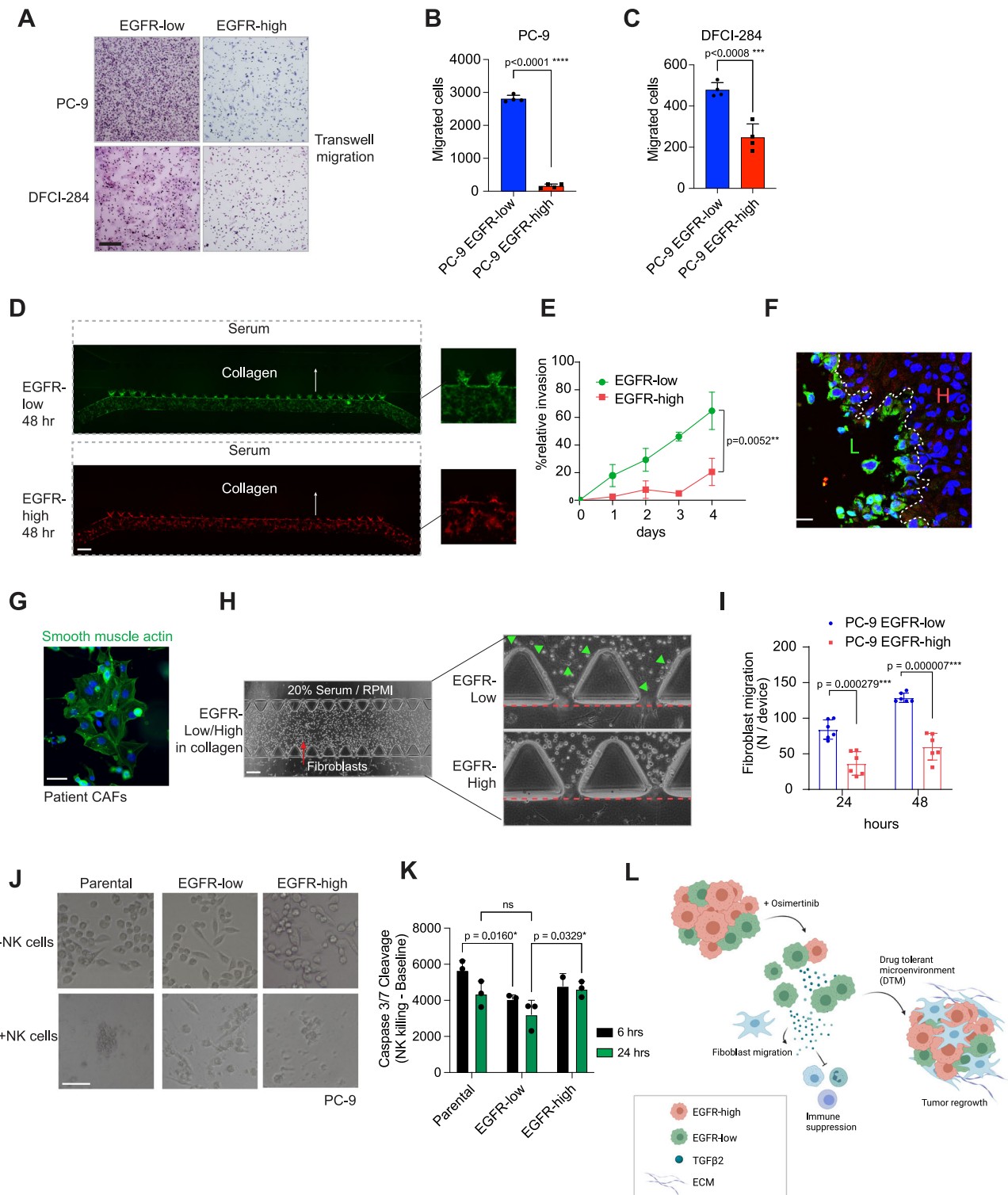

that both genetic and non-genetic factors contribute to treatment outcomes.

Microenvironmental factors can impact gene expression profiles that are regulated by epigenetic events[32]. The regional tumor heterogeneity also observed in this study could be the consequence of differential local microenvironments in the tumor, and can be influenced by cell-cell interactions, ECM interactions, or nutrient and oxygen status of the cells[32]. Our study reveals that in *EGFR*-mutant NSCLC, a minor EGFR-low expressing tumor clone can be essential to the development of drug resistance over time by microenvironmental modifications potentiating the tumor regrowth. Observed EGFR-low

expressing cells showed more invasive properties and modified the tumor microenvironment to potentiate cancer cell reoccurrence, indicating that protein expression heterogeneity can play an essential role in the development of drug resistance.

Our study demonstrated that EGFR-low expressing cells can be turned more sensitive to EGFR TKI treatment by epigenetic modulation. Although some tumors were able to regrow after EGFR TKI treatment regardless of the HDAC inhibitor co-treatment, the combination was able to prevent tumor regrowth more efficiently than the single-agent drugs. As seen in lung cancer patients, multiple mechanisms of resistance can influence drug resistance, and in many patients,

**Fig. 5 | EGFR-low cells are more invasive and modulate the tumor micro-environment. A** EGFR-low cells show enhanced motility in a transwell migration assay. Migrated cells were stained with hematoxylin. Scale bar is 100 µm. **B**, **C** Quantification of the transwell migration assay. $N = 3$ biologically independent experiments. **D** Microfluidic invasion assay. PC-9 EGFR-low or EGFR-high cells were seeded into the proximal side channel of the microfluidic chip, and their invasion towards collagen was measured over time. Serum was added to the proximal side channel to attract the cells. Scale bar is 1000 µm. **E** Quantification of the microfluidic invasion assay. The y-axis indicates EGFR-low and EGFR-high cell invasion area within collagen in the middle channel. $N = 3$ biologically independent experiments. **F** EGFR-low cells reside in the invasive front of the PC-9 xenograft tumors. Distribution of EGFR-low vs EGFR-high cells in 1:1 grafted tumors. L: EGFR-low, H: EGFR-high. Scale bar is 50 µm. **G** SMA expression in patient tumor-derived CAFs. Scale bar is 30 µm. **H** Microfluidic fibroblast migration assay. Patient CAFs were seeded into the proximal side channel, and PC-9 EGFR-low or EGFR-high cells were seeded into the middle chamber. Serum was added to the proximal side channel to attract the cells. Fibroblasts that migrated into the channel are highlighted with green triangles. Scale bar is 1000 µm. **I** EGFR-low cells attract more fibroblasts. Quantification of the CAF migration assay. $N = 3$ technical replicates, representative experiment of 6 microfluidic chips. **J** NK cell killing in PC-9 parental, EGFR-low and EGFR-high cell lines. Scale bar is 100 µm. **K** EGFR-low cells are less sensitive to NK cell killing in two different time points. $N = 3$ biologically independent experiments. **L** Schematic of osimertinib resistance formation over time. EGFR-low cells are intrinsically more tolerant to EGFR inhibitor and can survive the initial drug insult better than EGFR-high cells. Surviving EGFR-low cells secrete more TGFβ, attract fibroblasts to the microenvironment and are more resistant to immune cell killing, enabling the regrowth of all the remaining tumor cells. Data in (**B**, **C**) are presented as mean ± SEM and as mean ± SD in (**E**, **I**, **K**). Data were analyzed by an unpaired student's $t$-test in (**B**, **C**), by a paired student's $t$-test in (**I**), by two-way ANOVA in (**E**), and by two-way ANOVA, Tukey's test in (**K**).

---

a genetic resistance mechanism cannot be detected[2]. Our results provide an additional non-mutational mechanism for EGFR TKI resistance that can play an important role in the MRD state and thus significantly contribute to the treatment outcome.

Overall, our findings suggest that protein level heterogeneity may be an important contributor to treatment outcomes, especially during the early days of drug resistance, while observable genetic resistance mutations may take longer to develop. Therefore, combination therapies targeting both genetic and non-genetic factors may be a more effective approach to prevent or overcome drug resistance in *EGFR*-mutant NSCLC. Further studies are needed to elucidate the complex interactions between heterogeneous tumor cells and their microenvironment and to develop more effective therapeutic strategies that take into account the heterogeneity and plasticity of cancer cells.

## Methods

This study complies with all relevant ethical regulations. For treatment-naiive tumors, normal lung tissue and EGFRwt tumors used for histochemistry: A statement of approval for the study protocol was given by the Ethics Committee of the Hospital District of Helsinki and Uusimaa (HUS/970/2021) and research permit is from Helsinki University Hospital (HUS/237/2021). For PDX Studies: Patient derived xenografts were generated from tumor biopsies or pleural effusions from EGFR mutant patients undergoing clinical biopsies and propagated in mice. All patients provided written informed consent. The study was conducted in accordance with the Declaration of Helsinki and was approved by the Dana Farber Cancer Institute.

### Cell culture & reagents

Cell lines were obtained from ATCC and cultured in RPMI medium (Gibco) with 10% FBS (Sigma) and 1% penicillin / streptomycin (Life Technologies). DFCI-284 was established by using a previously published protocol[33]. Osimertinib, Vorinostat and Panobinostat for cell culture experiments were purchased from Selleck (S7297, S1047, S1030).

### Incucyte cell confluency assay

Cells were seeded in a 96-well plate, 1000cells/well, and on the following days the cells were treated using HP D300e digital dispenser. Imaging and analysis were performed using Incucyte Zoom / S3 imagers (Essen Biosciences).

### 3D cell confluency assays

Cells were seeded into Matrigel and allowed to form spheroids overnight. The next day, cells were treated with drugs and imaged every 8–12 h using a Cytation 5/Biospa 8 automated incubator (Biotek). CellEvent Caspase 3/7 green apoptosis detection reagent (C10423, Invitrogen) was added to the cell culture medium when apoptosis was monitored.

### Cell titer glo assay

Cell titer glo (G77570, Promega) was conducted according to manufacturer's instructions.

### Cytokine profiling

Cytokine profiling was performed from the cell culture medium after 48 h of culture. The profiling was performed using Bio-Plex Pro Human Cytokine 27 panel (#M500KCAYF0Y, Biorad) using Luminex instrumentation.

### CRISPR knockout of FN1, ITGB3 and TGFb2 in PC9 EGFR-low cells

The *FN1*, *ITGB3*, and *TFGFB2* KO cell lines were generated by the Finnish Genome Editing Center (FinGEEC, HiLIFE and the Faculty of Medicine, Research Programs Unit, University of Helsinki, and Biocenter Finland). Knockouts were validated using western blot. More detailed CRISPR KO and validation protocols (incl. sequence information) are included in the Supplementary Methods.

### Data analysis

When two groups were compared, two-tailed unpaired $t$-test was used to calculate significance. One-way ANOVA was used when comparing 3 or more groups, and Tukey's multiple comparisons test was used for post hoc analysis. The statistical tests used are indicated in the figure legends. Data are presented as mean -/+ stdev or sem, indicated in the figure legends. $p \leq 0.05$ was considered significant, and all assays were repeated using 3-6 biological replicates. In this study, biological replicates refer to independent experiments conducted at different time points while technical replicates involve repeated measurements from different flasks of the same cell line, conducted simultaneously. GraphPad Prism 9.4 software was used for statistical analyses. Experiments shown in Fig. 1A, C, D, Fig. 4F, and Fig. 7H were conducted once using patient samples or individual PDXs. Experiments were repeated three times in (Fig. 3J, K) with similar results. Experiments were repeated two times in (Fig. 4K), two-three times in (Fig. 6B–D), once for the validation markers in (Fig. 6I, J), and two times in (Fig. 6K) with similar results.

### Drug sensitivity scoring

The pharmacogenomics dataset Genomics of Drug Sensitivity in Cancer (GDSC)[34] was used in our analysis. The CGP dataset contains mRNA expression profiles and pharmacological responses for hundreds of anti-cancer drugs. In this dataset, the drug response is represented by the natural logarithm of the IC50 value, which corresponds to the half maximal inhibitory concentration of an anti-cancer drug. Twenty-three lung cancer cell lines with EGFR mutant in CGP dataset were used to observe the correlation between EGFR expression and drug sensitivity to seven EGFR inhibitors, including Afatinib (2 replicated experiments), Cetuximab, Gefitinib, Pelitinib, AZD3759, Osimertinib, and Sapitinib (Supplementary Data 2).

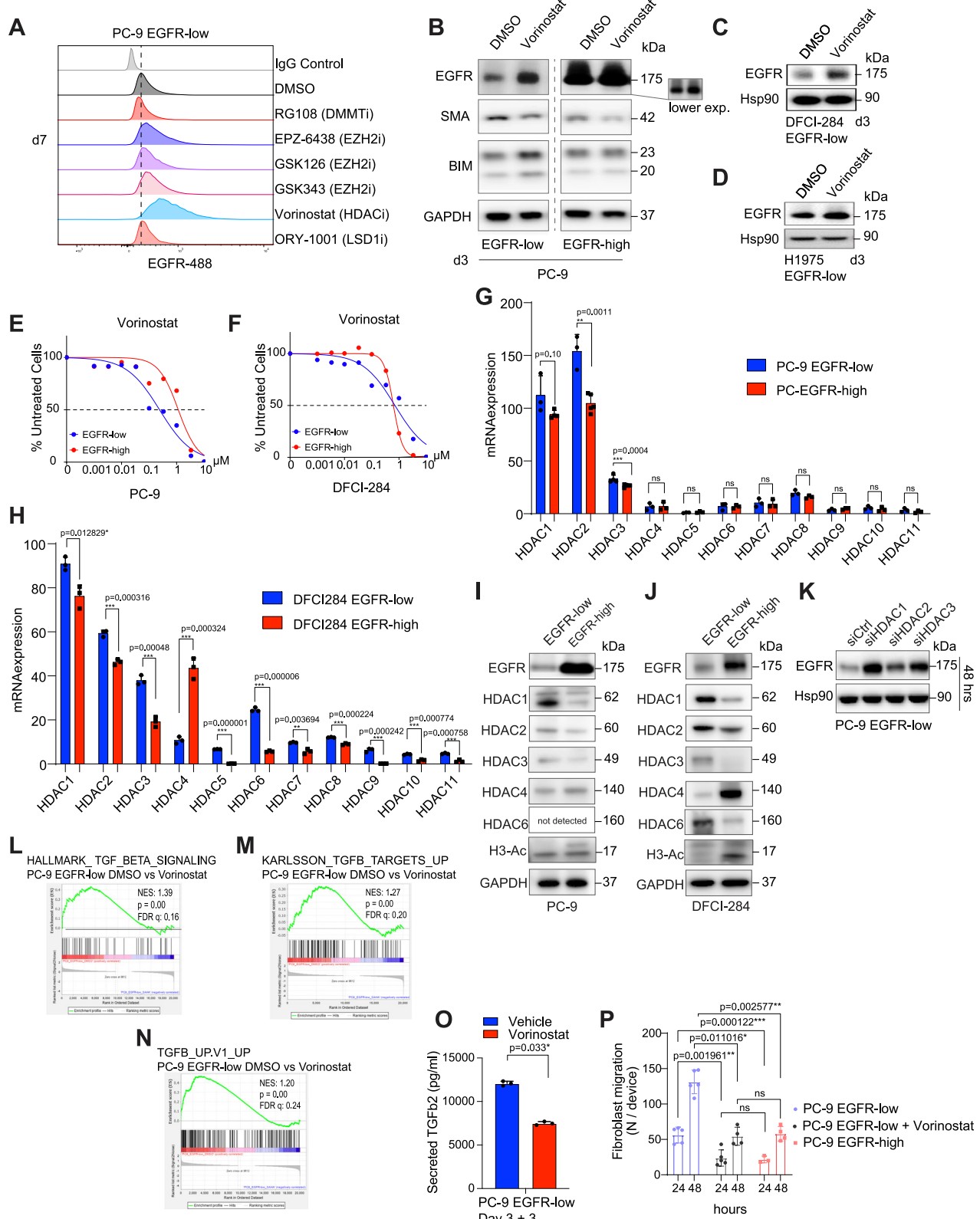

To analyze EGFR mutant NSCLC cell lines, we divided them into two groups based on their EGFR expression levels (high vs. low, threshold 4.3). We employed the student's t-test to compare the IC50 values of EGFR inhibitors between the two groups. Differential expression analysis was performed using DESeq2[35], which identified differentially expressed genes (DEGs) between the two groups. Additionally, we used Gene Set Enrichment Analysis (GSEA)[35] to determine the enriched gene sets, including those from KEGG, Gene Ontology, Cancer hallmarks, and CGP (chemical and genetic perturbations) databases. We ranked all genes by their statistical significance of differential expression and used this pre-ranked list to identify the enriched gene sets. Chosen clinically relevant mutations were EGFR p.E756_A750del ELREA, L858R, T790M).

**Fig. 6 | EGFR expression can be modulated using epigenetic inhibitors.**
**A** Epigenetic modulator screen for EGFR expression. Cells were treated with the modulators for 7 days, after which the membrane EGFR was measured in viable single cells. IC50 value of each drug was used. IgG control was used as a negative control. **B** Vorinostat upregulates EGFR, reduces SMA and induces BIM expression. GAPDH was used as a loading control. **C, D** EGFR expression is induced by vorinostat also in DFCI-284/H1975 EGFR-low cells. Hsp90 was used as a loading control. **E, F** EGFR-low cells are more sensitive to vorinostat treatment. **G, H** mRNA of different HDACs in PC-9/DFCI-284 EGFR-low vs EGFR-high cells. $N$ = 3 biologically independent experiments. **I, J** HDAC protein expression in EGFR-low vs EGFR-high cell lines. H3-Ac marks overall H3 acetylation in the cells. GAPDH was used as a loading control. **K** Knock down of HDACs induces robust EGFR activation. Western

blot analysis in PC-9 EGFR-low cells. Hsp90 was used as a loading control. (**L-N**) TGFβ signatures in PC-9 EGFR-low cells with or without 1 μM vorinostat treatment. **O** Vorinostat decreases the secretion of TGFβ2. TGFβ2 was measured from the cell culture medium after Cells were treated for 3 days with vehicle DMSO or 1 μM vorinostat, after which the medium was changed. TGFβ2 was measured after 3 days from the medium change. $N$ = 3 biologically independent experiments. **P** Vorinostat reverses the fibroblast attraction induced by EGFR-low cells. Data represent three individual experiments. Each experiment was repeated 3 times with 5 microfluidic chips in all conditions except for 3 chips in PC9 EGFR-high 24 h due to Matrigel disruption. Data in (**G, H, O, P**) are presented as mean ± SD. Data were analyzed by an unpaired student's $t$-test in (**G, H, P**), by a paired student's $t$-test in (**O**).

## Drug testing in tumor-on-a-chip
Microfluidic chips by MIMETAS (6405-400-B, MIMETAS) were used to study drug responses of patient derived organoids on a chip. Middle channel was seeded with 2 μl suspension of tumor cells aggregates ($2 \times 10^3$ cell/ μl) in 80% collagen I (4 mg/ml) (344702001, Bio-Techne) followed by incubation at +37 °C for 15 min to allow gel solidification. Distant side channel was seeded with $17.5 \times 10^3$ HUVEC cells in endothelial media and the chips were inverted at a 75 °C angle for 3 hours to allow endothelial cells attachment to the ECM. Second distant side channel was supplemented with organoid media to support their formation. The devices were then incubated at +37 °C for three days to allow organoid formation and endothelial tubule establishment. After three days, 40 nM osimertinib, 1 μM vorinostat, 20 nM panobinostat, and their combinations were injected into the endothelial tubule and incubated at +37 °C for 72 h. The chips were then stained Live/Dead fluorescence staining was performed by loading 1:5 AO/PI Staining Solution (CS2-0106-5ML, Nexcelom ViaStain™) diluted in PBS followed by washing with PBS. Following the staining, the chips were imaged with Nikon Eclipse 80i fluorescence microscope and images were quantified using ImageJ.

## EGFR expression in healthy lung scRNA-seq data
UMAP visualization and the accompanied cell type labels and EGFR gene expression values were downloaded from The Human Protein Atlas. The expression values shown in the UMAP visualization are nTPM (transcripts per million) values.

## Ectopic degradable EGFR
Mutant EGFR (pDONR233_EGFR_p.ELREA_746del, Addgene #82911) was cloned into pLEX_305_C'dTAG (kindly provided by Prof. Behnam Nabet[35],) lentiviral plasmid by using Gateway cloning, obtaining mutEGFR-dTAG plasmid. Cloned plasmids were verified by sanger sequencing. In the functional version of the plasmid, a spontaneously generated stop codon between the mutEGFR and degradation part (dTAG) was removed by using QuikChange site directed mutagenesis kit (Agilent) to allow full construct read-through. In the "non-functional" version of the plasmid, the stop codon was left in between the mutEGFR and the dTAG part of the plasmid, providing a negative control plasmid for degradation and only encoding for mutEGFR. C'GFP-dTAG plasmid was used as an additional control for the assays.

## Flow cytometry for cultured cell lines
Cells were collected using Accutase (A1110501, Thermo Fisher Scientific), and counted with Countess cell counter (Invitrogen). Suspension confluency was adjusted to $1 \times 10^6$ cells/ml, after which the cells were stained with 1:100 zombie viability dye for 20 min in the dark at RT. Cells were spun down and washed with PBS, followed by 30 min staining with conjugated antibodies in flow buffer (10 % FBS-PBS), on ice and in the dark. Samples were spin washed 3 times, resuspended into flow buffer and acquired using Fortessa analyzer (BD Biosciences). The data were analyzed with FlowJo version 10.8 (BD Biosciences). Cell sorting was conducted using a BD Melody instrument.

## Flow cytometry for patient-derived xenograft tumors
Patient derived xenograft tumors were collected and minced with a scalpel, after which they were shaken at 140 RPM in 0.2% collagenase A and cell culture medium for 3-4 h, +37 °C. The cell suspension was treated with TrypLe Express solution (12604013, Thermo Fisher Scientific) for 20 min at +37 °C to obtain a single cell suspension. The cell suspension was adjusted to $1 \times 10^6$ cells / ml and stained with zombie viability dye (BioLegend) for 20 min at RT in the dark. Cells were spun down and washed with PBS, followed by 30 min staining with conjugated antibodies in flow buffer (10% FBS-PBS), on ice and in the dark. Samples were spin washed 3 times, resuspended into flow buffer and acquired using Fortessa analyzer (BD). The data were analyzed with FlowJo version 10.8 (BD Biosciences).

## Fibroblast migration assay in a microfluidic device
Labelled tumor cells were counted and seeded into the middle chamber of the microfluidic device (DAX01, Sigma Aldrich) in collagen matrix (11563550, Corning). After collagen polymerization, patient-tumor derived fibroblasts were counted and spin washed with PBS, and patient tumor-derived fibroblasts were added to the left side channel. Finally, 20% serum-RPMI was added to the right-side channel. Migrated fibroblasts were counted under a microscope after 24 and 48 h of cell seeding. Migrated fibroblasts were identified by cell morphology, fibroblast marker staining, and negativity for GFP/mCherry (in the labelled tumor cells). Drug treated tumor cells were seeded and treated for 4 days before counting and loading into the microfluidic device.

## Gene set enrichment analysis
Gene Set Enrichment Analysis (GSEA) was performed using the GSEA software (v4.3.2) to examine pathway analysis for the differential expression analysis results between the EGFR-low and EGFR-high subsets. The test statistic utilized for ranking the genes was defined as the -log10 of the nominal $p$-value, multiplied by the sign of the fold-change. Genes with duplicated rankings were removed as per the documentation. Preranked GSEA was executed with default settings. The gene set database c2.all.v2023.2.Hs.symbols, comprising curated gene sets from online pathway databases, was employed for this analysis.

## Immunofluorescence
Cells grown and treated on coverslips were fixed with 4% PFA and permeabilized with 0.1 % Triton X-100. 2D immunofluorescence stainings were performed using standard protocols. 3D cultured samples were fixed with 2% PFA, permeabilized with 0.25% Triton X-100, and blocked with 10% goat serum in PBS. Primary antibodies were diluted in blocking buffer and 3D cultures were stained overnight. After three washes with IF buffer (0.1 % BSA, 0.2 % Triton X-100, 0.05 % Tween-20 in PBS), the cells were incubated with secondary antibodies in blocking buffer and washed again. Nuclei were counterstained with 1 μg/ml DAPI (Cell Signaling Technology) and mounted with Immu-Mount reagent (FIS9990402, Fisher Scientific). Imaging was performed using Nikon Eclipse 80i and Leica SP5 confocal microscopes (Dana-Farber Cancer Institute Confocal and Light Microscopy Core

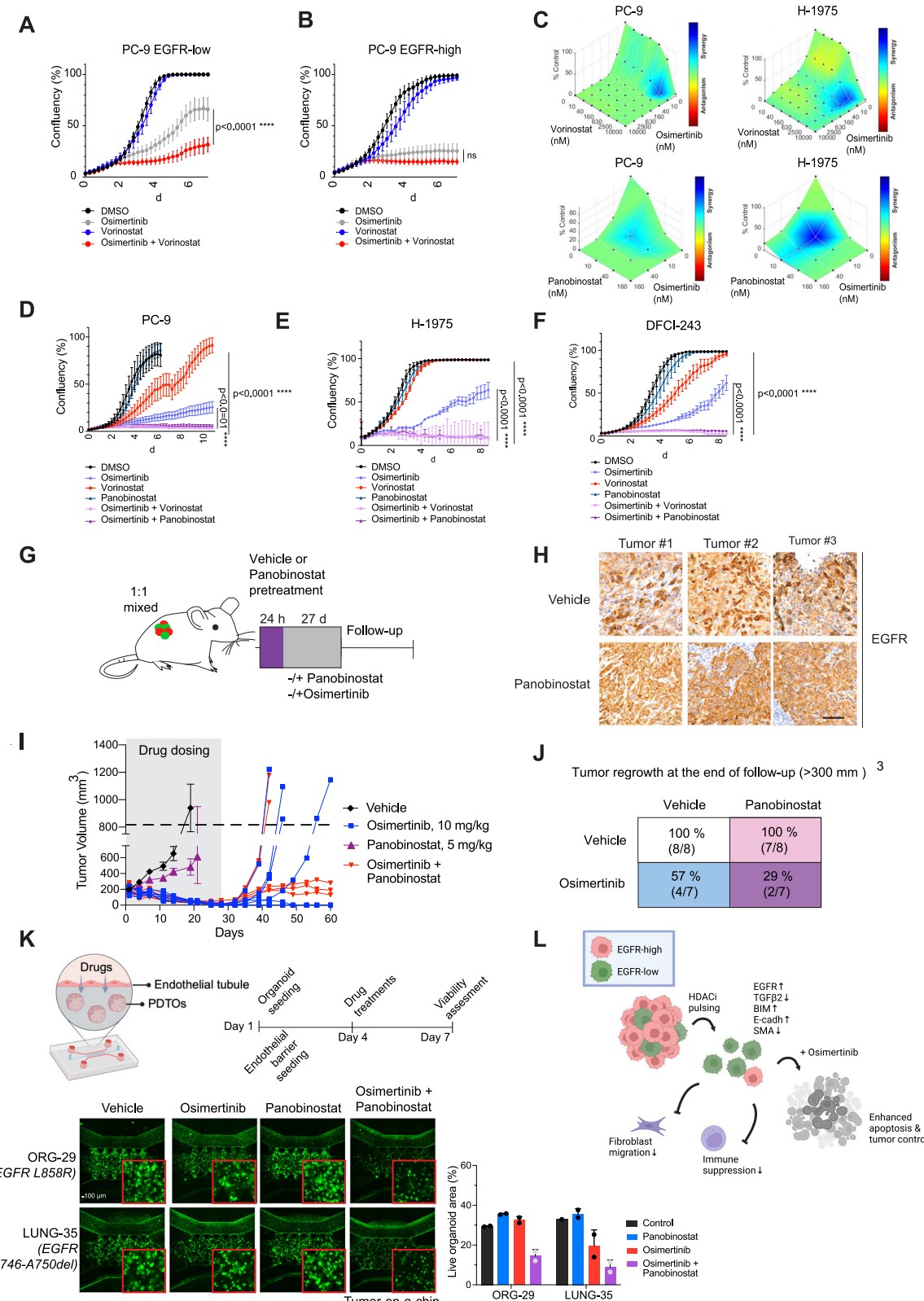

Facility). Quantifications for nuclear staining, Ki67, and pH2aX were performed using ImageJ software.

## Immunohistochemistry

A statement of approval for the study protocol was given by the Ethics Committee of the Hospital District of Helsinki and Uusimaa (HUS/970/2021) and research permit is from Helsinki University Hospital (HUS/237/2021). The patient participation was voluntary and required an informed consent. *EGFR* mutation has been detected in a diagnostic NGS in-house cancer gene panel (*ALK, EGFR, KEAP1, KIT, KRAS, MET, NRAS, PDGFRA, PIK3CA and STK11*).

Tissues were collected and fixed with 4% paraformaldehyde (PFA) and embedded in paraffin. 5 μm sections were cut and the slides were deparaffinized in 1X Antigen retrieval citrate buffer solution (Dako) in +70 °C overnight. Sections were stained using EGFR antibody (MA5-16360, Life Technologies). Staining was detected with DAB and

**Fig. 7 | Pharmacological upregulation of EGFR sensitizes cells to osimertinib and prohibits tumor regrowth. A, B** Vorinostat synergizes with osimertinib. PC-9 EGFR-low vs EGFR-high cells were treated with either vehicle, 10 nM osimertinib, 1 uM vorinostat or the combination of the two drugs. **C** Panobinostat synergizes with osimertinib in a lower concentration. Cells were treated with drugs and synergy assay was performed after 72 h of drug treatment. Blue color indicates drug synergy. **D–F** HDAC inhibitors prevent the regrowth of osimertinib-treated cells. Cell lines were treated with either vehicle, 10 nM osimertinib, 1 uM vorinostat, 50 nM panobinostat, or with combination of osimertinib + vorinostat or osimertinib + panobinostat. **G** Mouse treatment scheme. 1:1 mixed PC-9 EGFR-low:EGFR-high cells were mixed and grafted into xenograft mice. After the tumors were formed, the mice were first pre-treated with either Vehicle or Panobinostat. After the pre-treatment the mice were received either vehicle, panobinostat, osimertinib or the combination of the two drugs for 21 days. Follow-up was until day 60. **H** Panobinostat increases EGFR expression in vivo treated tumors. Representative tumor images shown from both treatment groups. Scale bar is 100 μm. **I, J** Panobinostat reduces tumor regrowth. Tumor growth was followed-up until day 60. Tumors bigger than 300 mm2 were considered as regrown. **J** Schematic how HDAC inhibition prevents osimertinib resistance formation over time. HDAC

inhibitor can induce higher expression of EGFR in the EGFR-low cells, thus making them more sensitive for osimertinib treatment. This can lower the secretion of TGFβ, fibroblasts attraction and immune suppression induced by the EGFR-low cells and prevent or delay tumor regrowth. **K** Combination drug testing on a tumor-on-a-chip (TOC) model. Two EGFR mutant patient-derived organoid models were integrated into the TOC together with endothelial tubule to test drug permeability. After the formation of the tubule and the tumor niche, 40 nM osimertinib, 1 uM vorinostat, 20 nM panobinostat, and their combinations were injected into the endothelial tubule and incubated for 72 h. Representative pictures of the chips are shown. The quantification graph illustrates the % coverage of live organoids within the tumor area of the chip. $N = 2$ technical replicates. **L** Therapeutic strategy to eradicate EGFR-low cells. Treatment with an HDAC inhibitor induces a phenotypic switch characterized by increased EGFR expression and a decreased EMT-like state, leading to reduced fibroblast attraction and immune suppression. Consequently, the combination of the HDAC inhibitor with osimertinib enhances tumor cell killing and diminishes tumor regrowth. Data in (**A, B, D, E, F, K**) are presented as mean ± SD. Data were analyzed by two-way ANOVA in (**A, B, D, E, F**). $N = 3$ biologically independent experiments in (**A, B, D, E, F**).

counterstained with Hematoxylin. Histochemical stainings were carried out using standard techniques. Images were taken with a Leica DM LB microscope and Studio-Lite 1.0 software (Biomedicum Imaging Unit, University of Helsinki). Images and slides were blinded for analysis.

### Incucyte cell confluency assay
Cells were seeded in a 96-well plate, 1000cells/well, and on the following days the cells were treated using HP D300e digital dispenser. Imaging and analysis were performed using Incucyte Zoom / S3 imagers (Essen Biosciences).

### Invasion assay in microfluidic device
Microfluidic chips by AIM Biotech were used to study the collagen invasion of cancer cells (DAX01, Sigma Aldrich). Middle chamber was seeded with 80 % collagen (11563550, Corning). Distant side channel was seeded with 20% serum-PBS, while proximal channel was seeded with 10,000 labelled cancer cells in their normal culture medium. The chips were imaged daily with Nikon Eclipse 80i fluorescence microscope equipped with Z-stack (Prior) and CoolSNAP CCD camcer (Roper Scientific) and invasion was quantified using NIS Elements AR software package.

### mRNA sequencing and gene-set enrichment analysis
RNA was isolated using the RNeasy Plus Kit (74034, Qiagen), after which the samples were sequenced and processed in the Molecular Biology Core Facilities (MBCF) in Dana-Farber Cancer Institute. Gene set enrichment analysis was performed using GSEA software (UC San Diego & Broad Institute).

### NK cell killing assay
NK cells were expanded from PB mononuclear cells using K562-mbIL21-41BBL feeder cells as previously described[36]. Expanded purified NK cells were frozen and before the use, they were thawed in a waterbath and mixed with NK cell medium (RPMI 1640 supplemented with 1% Penicillin/Streptomycin, 1% Ultraglutamine, and 10% FBS) and centrifuged (500 x $g$, 5 min at RT). The supernatant was then aspirated, and the pellet was resuspended with 5 ml thawing media +25 U/ml benzonase (70746-3, Merck) to prevent cell clumping due to DNA released by dead cells and incubated at 37 °C for 15 min. This was followed by adding 10 ml thawing media and centrifugation (500 x $g$, 5 min at RT). The pellet was resuspended in NK media at $2 \times 10^6$ cells/ml + 10 ng/ml IL-2 and incubated at 37 °C overnight for recovery. On the next day, recovered NK cells were washed and resuspended in NK cell medium + 10 ng/ml IL-2. To assess NK cell killing, we co-cultured NK cells with the tumor cells in 2:1 ratio, respectively. After 6 and 24 h the killing effect was measured using Caspase 3/7 Glo (Promega). Luminescence was measured using Spark multimode microplate reader.

### siRNA knock downs
For the siRNA knock downs *FN1*, *TGFB2*, and *ITGB3* were targeted in the PC9 EGFR-low cancer cells with 0.25-1 μg siRNA for 24 h in transfection media and incubated at 37 °C (sc-29315, EHU084101, sc-29375, Santa Cruz and Merck). Knock down was verified by using western blot. After the knock downs the invasion assay was performed as above. The chips were imaged daily with Nikon Eclipse 80i fluorescence microscope and invasion was quantified using ImageJ.

### TGFb2 ELISA assay
Medium was collected after 3 days of cell culture and TGFβ2 ELISA was conducted using manufacturer's instructions (R&D Systems # DB250).

### Tumor organoid establishment and culture
Primary lung tumor surgical samples were collected under Helsinki University Hospital (HUS) IRB (HUS/8/2022) with a statement from the institutional ethical board HUS/970/2021. Surgical tissues were delivered to the laboratory within an hour since the tissue was collected. The tissue was minced using surgical scalpels to 1-2 mm³ fragments and further dissociated with collagenase-medium on a shaker at +37 °C overnight. The digestion was stopped by one wash with 20% fetal bovine serum (FBS) in cold DMEM and followed by two washes with PBS. We embedded the cells/fragments into 3D culture using Matrigel. 30 minutes after embedding Matrigel solidified and organoid medium was added. Based on their growth, organoids were passaged once every 1-2 weeks at a 1:2 split ratio. Organoids were dissociated with pre-warmed TrypLE express for 5–10 min at +37 °C. To stop TrypLE effect, cold 20% FBS in DMEM/F12 was added. The presence of tumor-bearing driver mutations in the organoids was validated using whole-exome sequencing.

### Western blot
Cells were plated at a density of 300,000 per well in a 6-well plate and samples were lysed the following day after seeding by RIPA lysis and extraction buffer (Thermo Fisher, 89900) supplemented with Halt™ Protease Inhibitor Cocktail (Thermo Fisher, 78430). The cells were scraped, collected, and lysed on ice for 10 min. Lysates were centrifuged for 15 min at 4 °C, and the supernatants were collected and kept on ice. Protein concentration was determined using the Pierce™ BCA Protein Assay Kit (Thermo Fisher, 23227), following the manufacturer's protocol. Samples containing 30 μg of protein were prepared by adding 4x loading buffer, diluted to a final volume of 20 μL, and heated at 95 °C. After cooling to room temperature, 5 μL of PageRuler Prestained Protein Ladder (Thermo Fisher, 11812124) was loaded, alongside 20 μL of sample per well, into 4–20% Mini-PROTEAN TGX Stain-Free Gels (Bio-Rad, 4568096). Electrophoresis was

performed at 80 V, followed by transfer using the Trans-Blot Turbo RTA Transfer Kit (Bio-Rad, 1704270) and Trans-Blot® Turbo™ Transfer System (Bio-Rad, 1704150). Membranes were briefly washed with MQ water and blocked with a buffer containing 1x TBS, 0.05% Tween-20, and 5% nonfat dry milk for 1 h at room temperature on a shaker. Primary antibodies (1:1000 dilution in blocking buffer) were incubated overnight at 4 °C. The next day, membranes were washed once for 15 min and three times for 5 min with TBS-T (1x TBS, 0.05% Tween-20). Membranes were incubated with secondary antibodies (1:10,000 dilution in blocking buffer) for 1 h at room temperature, followed by washes as previously described. Detection was performed using Clarity Western ECL Substrate (Bio-Rad, 1705061), and membranes were imaged using the Azure™ Imaging System (Azure Biosystems).

### Xenograft studies

Patient-derived xenografts were generated from tumor biopsies or pleural effusions from *EGFR* mutant patients undergoing clinical biopsies and propagated in NSG mice as described previously[34]. All patients provided written informed consent. All animal studies were conducted at Dana-Farber Cancer Institute with the approval of the Institutional Animal Care and Use Committee in an AAALAC accredited vivarium. For PC9 xenograft model, female NCr nude mice, 6-weeks old were purchased from Taconic Bioscience, Inc. (Germantown, NY). Animals were acclimated for at least 5 days before initiation of study. PC9 cells (EGFR-low and EGFR-high in a 1:1 ratio) were implanted subcutaneously with 50% Matrigel (Corning, NY). Tumors were allowed to establish to 200 ± 50 mm³ in size before randomization using Studylog software (San Francisco, CA) into various treatment groups with 8–10 mice per group. No statistical methods were used to pre-determine sample sizes but our sample sizes are similar to those reported in previous publications[34]. During treatment, tumor measurements were taken using calipers and body weights monitored twice a week. Tumor volumes were calculated using the following formula: (mm³) = length × width × width × 0.5. Mice were immediately euthanized when tumor volume exceeded 2000 mm³ or if the tumors became necrotic or ulcerated. Data collection and analysis were not performed blind to the conditions of the experiments. The vehicles used for each compound were as follows: Osimertinib, 0.5% HPMC (hydroxy propyl methylcellulose) in water administered orally once daily; Panobinostat, 10% DMSO with 90% of 5% dextrose in water administered IP on a 5ON, 2OFF schedule.

### CRISPR KO cell lines

**TGFβ2 gRNA validation.** Two guide RNA constructs targeting TGFβ2 (Supplementary Data 2) cloned into 3rd generation Cas9 and a GFP protein expressing lentivirus vector (FinGEEC and GBU, University of Helsinki). The gRNA validation for the two guide RNAs was done in PC9 EGFR-low cells, and the functionality was assessed with Surveyor's assay (FinGEEC and GBU, University of Helsinki).

To amplify and check mutation in the DNA fragment within the guide RNA targeted genomic DNA region, two sets of primers were designed at FinGEEC (Supplementary Table 3). Correct sized PCR fragment (570 bp) with primer pair **2** were assessed with a 2% agarose gel and PCR products were purified for surveyor's assay.

**CRISPR/Cas9 knockout of FN1, ITGB3 and TGFβ2 in PC9 EGFR-low cells.** PC9 EGFR-low with knock-out (KO) of FN1, ITGβ3, and TGFβ2 as well as controls PC9 EGFR-low single and dual non-target cell lines were generated by FinGEEC via CRISPR/Cas9 as described in ref. 35. Cloned and distributed constructs by GBU used for cell line creations (Supplementary Tables 4, 5).

The FN1 KO cell line was generated with a previously validated single on-target guide RNA. The ITGB3 KO cell line was generated by

transfection of a dual on-target gRNA construct[35,37]. The TGFβ2 KO cell line was generated with co-transfection of two gRNAs. Control cell lines were generated with a single and a dual non-targeting gRNA construct each. TransIT-X2® Dynamic Delivery System (MIR 6003, Miros Bio) was used for all transfections. After 48 h of transfection, positive clones were sorted by flow cytometry based on PC9 EGFR-low GFP expression, then plated into 96-wells plate, one cell per well. For the monoclonal cell lines, the cells were incubated at 37 °C incubator with 5% $CO_2$ until confluent, then were expanded gradually.

### Reporting summary

Further information on research design is available in the Nature Portfolio Reporting Summary linked to this article.

## Data availability

Non-sensitive data generated or analyzed during this study, associated protocols and materials are provided by the corresponding author H.M.H. upon a request for research purposes. RNA sequencing data from the cell lines are deposited in Gene Expression Omnibus (GEO) under accession number GSE267515. The remaining data are available within the Article, Supplementary Information or Source Data file. Source data are provided with this paper.

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

## Acknowledgements

We want to thank Hideki Endoh for establishment of DFCI-169 cell line and Yoshisha Kobayashi for the copy number analysis. This work was funded by the National Cancer Institute grant R35CA220497 (P.A.J.). H.M.H. was funded by the Research Council of Finland, Sigrid-Jusélius Foundation, Cancer Foundation Finland, Nummela Foundation and Finnish Cultural Foundation. B.A. was funded by the Doctoral Programme in Clinical Research, University of Helsinki. C.A. was funded by grants from the Giovanni Armenise–Harvard Foundation and the Lung Cancer Research Foundation (LCRF). C.A. is supported by the Zanon di Valgiurata family through Justus s.s. B.N. acknowledges support from NCI K22 CA258805. Imaging was performed at the Biomedicum Imaging Unit, the University of Helsinki, supported by the Helsinki Institute of Life Science (HiLIFE) and Biocenter Finland.

## Author contributions

H.M.H., P.G., P.A.J., designed the study, analyzed and provided data, and provided funding. H.M.H. wrote the manuscript. B.A. performed experiments, analyzed and provided data and edited the manuscript. L.L., Ji.S., J.L., H.M., T.L., P.Ö.E., A.O., Y.E., T.T., N.B., C.A., H.D., B.N., M.J.P., B.K.E., performed experiments and analyzed the data. Jo.S., H.W., S.L. and D.B. analyzed and provided data. W.W.F. provided materials and edited the manuscript. E.S., K.B., S.M. and I.I. provided clinical data and patient samples.

## Competing interests

H.M.H. has been working within the past 3 years as a part-time Medical Advisor for Amgen AB. C.A. received research fees from Revolution Medicines, Aelin Therapeutics, Verastem, Roche and Boehringer-Ingelheim. P.A.J has consulting fees from AstraZeneca, Boehringer-Ingelheim, Pfizer, Roche/Genentech, Takeda Oncology, ACEA Biosciences, Eli Lilly and Company, Araxes Pharma, Ignyta, Mirati Therapeutics, Novartis, LOXO Oncology, Daiichi Sankyo, Sanofi Oncology, Voronoi, SFJ Pharmaceuticals, Takeda Oncology, Transcenta, Silicon Therapeutics, Syndax, Nuvalent, Bayer, Esai, Biocartis, Allorion Therapeutics, Accutar Biotech and Abbvie, Monte Rosa, Scorpion Therapeutics, Merus, Frontier Medicines, Hongyun Biotechnology and Duality; post-marketing royalties from DFCI owned intellectual property on EGFR mutations licensed to Lab Corp; sponsored research agreements with AstraZeneca, Daichi-Sankyo, PUMA, Boehringer Ingelheim, Eli Lilly and Company, Revolution Medicines and Astellas Pharmaceuticals; stock ownership in Gatekeeper Pharmaceuticals. B.N. is an inventor on patent applications related to the dTAG system (WO/2017/024318, WO/2017/024319, WO/2018/148440, WO/2018/148443, and WO/2020/146250). The Nabet laboratory receives or has received research funding from Mitsubishi Tanabe Pharma America, Inc. The remaining authors declare no competing interests.

## Additional information

[1]Translational Immunology Research Program (TRIMM), Research Programs Unit, Faculty of Medicine, University of Helsinki, Helsinki, Finland. [2]iCAN Digital Precision Cancer Medicine Flagship, Helsinki, Finland. [3]Department of Medical Oncology, Dana-Farber Cancer Institute, Boston, MA, USA. [4]Harvard Medical School, Boston, MA, USA. [5]Hematology Research Unit Helsinki, University of Helsinki and Helsinki University Hospital Comprehensive Cancer Center, Helsinki, Finland. [6]Department of Molecular Biotechnology and Health Sciences, Molecular Biotechnology Center, University of Torino, Torino, Italy. [7]School of Life Sciences and Technology, Tongji University, Shanghai, China. [8]Individualized Drug Therapy Research Program, Faculty of Medicine, University of Helsinki, Helsinki, Finland. [9]Department of Pulmonary Medicine, Heart and Lung Center, Helsinki University Hospital, Helsinki, Finland. [10]Department of General Thoracic and Esophageal Surgery, Heart and Lung Center, Helsinki University Hospital & University of Helsinki, Helsinki, Finland. [11]Human Biology Division, Fred Hutchinson Cancer Center, Seattle, WA, USA. [12]Department of Pharmacology, University of Washington, Seattle, WA, USA. [13]Department of Pathology, Helsinki University Hospital & University of Helsinki, Helsinki, Finland. [14]Experimental Therapeutics Core and Belfer Center for Applied Cancer Science, Dana-Farber Cancer Institute, Boston, MA, USA. [15]These authors contributed equally: Pasi A. Jänne, Heidi M. Haikala.
✉e-mail: heidi.haikala@helsinki.fi

