## [Peer Review file · Nature Communications]

Intratumor heterogeneity of EGFR expression mediates targeted therapy resistance and formation of drug tolerant microenvironment

Corresponding Author: Dr Heidi Haikala

Version 0:

Reviewer comments:

Reviewer #1

(Remarks to the Author)

The work by Alsaed and collaborators is an interesting analysis of tumor heterogeneity in the context of EGFR addicted lung cancer: they show the presence of EGFR low- and high-expressing cells in the context of the tumor, both in patient's samples and in lung cancer cell lines. EGFR low cells display an increased ability to invade, metastatize and drive the EMT phenotype through the production of TGbeta2.

Whereas the treatment with the EGFR-TKI osimertinib is effective on EGFR high-expressing cells, EGFR-low cells are more tolerant, thus allowing the development of a microenvironment that can promote the resistance also in EGFR-high cells.

Expression of EGFR depends strictly by post-transcriptional modifications and therefore authors look at the use of epigenetic modifiers as potential drugs to combine to EGFR-TKI in order to prevent or delay resistance.

The work is noteworthy and meet the expected standard, conclusions are supported by results, the methodology is clear and detailed.

Reviewer #2

(Remarks to the Author)

Authors demonstrated the heterogeneity of EGFR expression in EGFR mutated lung cancer, and clarified the mechanisms by which tumor cells expressed different levels of EGFR protein. They performed experiments using EGFR mutated lung cancer cell lines as well as clinical specimens, patient derived tumor cells, and patient derived xenografts. They showed EGFR-low-expressing tumor cells had EMT phenotype with high expression of integrin alphaV/beta3 and TGF-beta2, associated with high migrated activity and low sensitivity to NK cell killing. HDAC inhibitors enhanced EGFR protein expression and sensitized to EGFR-TKIs. The data shown are interesting, but there are several issues to be considered.

1. This reviewer could not see the legends for figures. Therefore, this reviewer could not follow detail of data shown in this paper.
2. Figure 2. There was no significant difference on levels of mutated EGFR DNA/RNA between EGFR-high expressing tumor cells and EGFR-low expressing tumor cells. What is the mechanism by which EGFR protein was inhibited in a population of EGFR mutated lung cancer cells? In "Figure 2 J", what does "Kobaysshi EGFR signaling PC-9 EGFR-high vs EGFR-low" mean?
3. Figure 4 F. EGFR-low tumor cells expressed higher levels of integrin alphaV/beta3 and TGF-beta2, compared with EGFR-high tumor cells. But it is unclear whether these molecules really induced mesenchymal phenotype of EGFR-low expressing tumor cells. Authors should knockdown these molecules in EGFR-low expressing tumor cells and show the results on EMT phenotypes such as EGFR expression and cell migration.
4. Figure 3H. It is unclear how EGFR-low expressing tumor cells support growth of EGFR-high expressing tumor cells.
5. Figure 6K. Does HDAC inhibitor treatment increase the expression of mutated EGFR protein in EGFR-low expressing tumor cells? This can be examined by western blot with the mutated EGFR specific antibody.

6. Figure 7H. Does HDAC inhibitor treatment increase the expression of mutated EGFR protein in EGFR-low expressing tumor cells in vivo?
7. Figure 7. Is combined treatment with HDAC inhibitor and EGFR-TKI effective against PDX of EGFR mutated lung cancer?

Reviewer #3

(Remarks to the Author)

Alsaed et al present a well-written manuscript concerning the heterogeneous expression level of mutant EGFR in lung cancer, and address the functional consequence of having a minor cell population with low mEGFR expression.

Overall, this is a good manuscript that is quite extensive from observation, to mechanism, to therapeutic intervention. However, on various key claims the authors remain rather correlative, while a more direct assessment is warranted. Below are a few major comments to improve the support for their claims.

1. It is of interest to understand the molecular mechanism why the mEGFR^{low} cells are resistant towards EGFR inhibition. In particular, how relevant is the EGFR-MAPK signaling pathway in tumor cells with low mEGFR expression (only assessment that is currently present is a EGFR gene signature)? The authors should characterize pathway components using biochemistry, including effects of inhibiting EGFR, pan-HER, pan-RAF, MEK or ERK? In parallel, perform dose-response measurements against pan-RAF, MEK and/or ERK inhibitors?
2. I am very intrigued by the origin of the EGFR^{low} cells. Can the authors include lung cancers in their analyses that are wild-type for EGFR, as well as normal lung epithelium? Are low EGFR expressors to be detected in these cancer subtype or normal lung epithelium as well? If so, do they share similarities with the low mEGFR cells as described in the manuscript?
3. In relation to 2. scRNA-seq on enriched populations of low versus high EGFR expressors from in vivo (PDX) tumors will provide superior in-depth insights into the phenotype of the cells as they appear in vivo, and the signaling pathways/ gene networks that are active in these cells (e.g. MAPK signaling and/or EMT).
4. The general assessment of EGFR expression towards TKI sensitivity is not informative. The point made by the authors is the existence of a minor cell population with low mEGFR expression. Expression level in these few cells is easily masked by expression levels in the dominant cell populations when bulk analyses are performed. Measuring the fraction of low EGFR cells within various cell populations might be more informative.
5. The part that centers around TGFbeta2 expression is interesting, but remains very much descriptive and correlative. Generation of TGFbeta2 knock-outs are required in order to pinpoint all those phenotypic observations to TGFbeta2. Same holds true for FN1.
6. Using epigenetic modifiers to restore mEGFR expression is interesting, but the downside is that pan-HDAC inhibitors have wide effects on all gene-expression networks and are thus not specific towards restoration of EGFR expression alone. Moreover, it seemed as if EGFR levels were boosted in both cell population (EGFR^{low} and high). For the conclusion that the cells become sensitive towards EGFR inhibition due to its pharmacological upregulation, it is required to restore/enhance mEGFR expression in a gene specific manner, e.g. using CRISPRa, in EGFR^{low} cells and show that this leads to same sensitivity towards Osimertinib when compared to the mEGFR high expressors. Currently, it cannot be excluded that the cells become sensitive towards EGFR inhibition due to the crude intervention into the cells epigenetic status and (potentially) accompanying induction of stress.

Minor comments

There are always pro's and con's about specificity of TKIs and which one is the best. By my knowledge, afatinib is not specific for EGFR (line 146). To avoid such discussions, can the authors include a whole family of EGFR and pan-HER inhibitors, including gefitinib, lapatinib, dacomitinib etc.

Typo in line 92

Version 1:

Reviewer comments:

Reviewer #2

(Remarks to the Author)

Authors appropriately revised MS.

Reviewer #3

(Remarks to the Author)

The authors performed many additional experiments and responded thoughtfully to the questions raised. Regarding the new experiments on ectopic expression of EGFR in the low populations. I am not sure whether I follow the conclusions towards osimertinib sensitivity. The effects seem very nuanced to me, for sure not as black and white as described.

Reviewer #4

(Remarks to the Author)

The paper by Alsaed includes an impressive amount of data that all together support important advances on the comprehension of the role of EGFR-low cell in TKI resistance in lung cancer. Moreover, a potential pharmaceutical strategy is suggested to sensitize these EGFR-low cells to TKI by epigenetic upregulation of EGFR.

Among others, the authors exploit also microfluidic approaches to study invasion, immune killing and drug responses. While conclusions appear overall correct, some methodological information are missing, in particular regarding the TOC experiments with PDOs (Fig. 7K). Even though another detailed method paper is under review, readers of this article should be able to understand the presented experiments.

How was the presence of the HUVEC endothelial barrier verified in the reported experiments? Do organoids expand in the TOC collagen over the 7 days? At what concentration (mg/ml) does 80% collagen correspond? Which green dye is used to stain live organoids? Please for clarity remove mention to drug distribution dynamics and interaction with 3D-ECM (lines 409-410), since it was not assessed in this work. Please, improve the paragraph 'Drug testing in tumor-on-a-chip' (lines 545-556). It is written that middle channel was seeded with tumor cells, but it should be with pre-formed organoids.

Fig. 5E, please define what is '%relative invasion' on y axis.

One of the paper conclusion is that EGFR-low cells exhibited an EMT phenotype. It would be informative to see pictures of EGFR-low cells showing a more mesenchymal phenotype than EGFR-low cells. Does the RNA sequencing of low versus high show upregulation of the typical EMT transcription factor signatures (Slug, Snail, Twist, Zeb)? The model of Fig. 7L (whose legend is missing) reports upregulation of E-cadh upon HDACi, but WB of Fig. 6B does not include E-cadh.

Version 2:

Reviewer comments:

Reviewer #4

(Remarks to the Author)

The authors have responded satisfactorily to my comments, and I fully support the publication of this interesting work.

Reviewer #1:

The work by Alsaed and collaborators is an interesting analysis of tumor heterogeneity in the context of EGFR addicted lung cancer: they show the presence of EGFR low- and high-expressing cells in the context of the tumor, both in patient's samples and in lung cancer cell lines. EGFR low cells display an increased ability to invade, metastatize and drive the EMT phenotype through the production of TGFbeta2. Whereas the treatment with the EGFR-TKI osimertinib is effective on EGFR high-expressing cells, EGFR-low cells are more tolerant, thus allowing the development of a microenviroment that can promote the resistance also in EGFR-high cells.

Expression of EGFR depends strictly by post-trascriptional modifications and therefore authors look at the use of epigenetic modifiers as potential drugs to combine to EGFR-TKI in order to prevent or delay resistance. The work is noteworthy and meet the expected standard, conclusions are supported by results, the methodology is clear and detailed.

Response: We would like to sincerely thank the reviewer for their thoughtful analysis and positive feedback on our work. There was no further comments to address from the reviewer #1.

Reviewer #2:

Authors demonstrated the heterogeneity of EGFR expression in EGFR mutated lung cancer, and clarified the mechanisms by which tumor cells expressed different levels of EGFR protein. They performed experiments using EGFR mutated lung cancer cell lines as well as clinical specimens, patient derived tumor cells, and patient derived xenografts. They showed EGFR-low-expressing tumor cells had EMT phenotype with high expression of integrin alphaV/beta3 and TGF-beta2, associated with high migrated activity and low sensitivity to NK cell killing. HDAC inhibitors enhanced EGFR protein expression and sensitized to EGFR-TKIs. The data shown are interesting, but there are several issues to be considered.

Comments:

1. This reviewer could not see the legends for figures. Therefore, this reviewer could not follow detail of data shown in this paper.

Response:

We apologize that the reviewer was not able to see the figure legends, we think the figure legend file might have not transferred and we failed to observe this mistake. We have now added the figure legends as a part of the new submission and are deeply sorry about this mistake.

2. *Figure 2. There was no significant difference on levels of mutated EGFR DNA/RNA between EGFR-high expressing tumor cells and EGFR-low expressing tumor cells. What is the mechanism by which EGFR protein was inhibited in a population of EGFR mutated lung cancer cells? In “Figure 2 J”, what does “Kobayashi EGFR signaling PC-9 EGFR-high vs EGFR-low” mean?*

Response:

Thank you for the valuable comment. It seems there might have been a misunderstanding regarding **Figure 2J**, possibly due to a previously missing figure legend, which we've now updated for clarity. Our intention was to show that, although no significant difference was found in DNA copy number or allelic frequency between mutEGFR low/high cells, a difference was observed in mRNA expression and RNA-level EGFR gene signatures (as illustrated in **Figure 2J**, ‘Kobayashi EGFR signaling’ is the name of the GSEA signature used to address this). Additionally, similar observations were made at the protein level (depicted in **Figure 2H** and **Supplementary Figures 2C-D**). These findings suggest that the observed effects are likely not due to differences in DNA itself but rather result from epigenetic or post-transcriptional regulation.

This insight prompted us to further investigate whether increasing EGFR expression in EGFR-low cells—through epigenetic means, specifically using HDAC inhibitors—could enhance their sensitivity to EGFR inhibitors. Our results support this approach. We hope this explanation helps clarify the role and findings associated with **Figure 2J** concerning EGFR gene expression signatures.

3. *Figure 4 F. EGFR-low tumor cells expressed higher levels of integrin alphaV/beta3 and TGF-beta2, compared with EGFR-high tumor cells. But it is unclear whether these molecules really induced mesenchymal phenotype of EGFR-low expressing tumor cells. Authors should knockdown these molecules in EGFR-low expressing tumor cells and show the results on EMT phenotypes such as EGFR expression and cell migration.*

Response:

We are very grateful for this valuable comment, we have now thoroughly addressed this question by performing new experiments both regarding EGFR expression, cell migration and sensitivity to EGFR-TKI (osimertinib) by using siRNA/CRISPR for the genes relevant for the mechanisms (*ITGB3*, *TGFβ2*, *FN1*).

We observed, that knock down of either of the three genes was able to reverse the migratory phenotype (new data added in **Supplementary Figure 5 A-C**, with the strongest phenotype reversal observed in *TGFβ2* KD followed by *ITGB3*), and KD/KO of the genes was able to better sensitize the EGFR-low cells to osimertinib (as shown in the new **Figure 4N**). Increased EGFR expression was associated with KD of all three genes, but the phenotype reversal was again most strong with *TGFβ2* KD (now shown in **Supplementary Figure 5H-I**).

Supplementary Figure 5A-B

Figure 4N

Supplementary Figure 4H-I

4. Figure 3H. It is unclear how EGFR-low expressing tumor cells support growth of EGFR-high expressing tumor cells.

Response:

We are grateful for this comment. One of our findings earlier was that EGFR-low cells are having different cytokine expression profile compared to EGFR-high expressing cells, including secretion of TGFβ2 (Figure 4I) and other secreted cytokines (**Supplementary Figure 5F**). In general, TGFβ isoforms are known to have a pleiotropic and context-dependent roles in the cell survival and proliferation (1). To specifically investigate the influence of EGFR-low cells on the proliferation of EGFR-high cells, we assessed the direct impact of adding recombinant TGFβ2 to the culture medium of PC-9 cells. Interestingly, this addition alone accelerated cell proliferation, suggesting that TGFβ2 could be at least partially responsible for the observed growth-supportive effects of EGFR-low cells. While other mechanisms involving different cytokines and differential gene expression patterns may also contribute (and we would like to study this further), the role of TGFβ2 appears significant. We have included these new findings in **Supplementary Figure 5F**.

Supplementary Figure 5F

5. Does HDAC inhibitor treatment increase the expression of mutated EGFR protein in EGFR-low expressing tumor cells? This can be examined by western blot with the mutated EGFR specific antibody.

Response:

We are grateful for this valuable comment, to address it we have explored the impact of HDAC inhibitors to mutant-EGFR expression in PC-9 cells with EGFR E746-A750del (exon19del) mutation, with a mutant-specific antibody (EGF Receptor E746-A750del Specific (D6B6) XP, #2085, CST). Besides the already pre-existing data in the previous version of the manuscript (the data in **Supplementary Figure 6B**), we observed a similar upregulation in mutant EGFR (as with total EGFR) concluding that both are altered by the HDAC inhibitors. New data regarding mutEGFR upregulation has now been added to **Supplementary Figure 6C**.

6. Figure 7H. Does HDAC inhibitor treatment increase the expression of mutated EGFR protein in EGFR-low expressing tumor cells in vivo?

Response:

We thank the reviewer for this insightful comment. To address this, we have examined the expression of EGFR mutated protein in PC-9 xenografts (1:1 mixed mutEGFR-low : mutEGFR-high) comparing the protein expression in vehicle or panobinostat treated mice. To our excitement, we could clearly observe that while in the vehicle-treated tumors the EGFR expression was heterogeneous, in the HDACi (panobinostat) treated group there was a uniform pattern of high EGFR expression observed, supporting our hypothesis that the effect observed earlier *in vitro* can contribute to the drug efficacy also *in vivo*. New data has been added to **Figure 7H**.

7. Figure 7. Is combined treatment with HDAC inhibitor and EGFR-TKI effective against PDX of EGFR mutated lung cancer?

Response:

We thank the reviewer for this comment. To address this, we conducted a new *in vivo* mouse experiment using a PDX model (DFCI-243) treated with osimertinib and panobinostat (HDACi). Even though a nice synergy was observed in the same cell line *ex vivo* (**Figure 7F**), the mouse model generated *in vivo* was highly sensitive to the previously used *in vivo* dose of osimertinib, and it was difficult to observe any synergistic effects using this model. Interestingly, in the osimertinib group we did observe in general more internal variation in the regrowth volumes in individual mice (data below, only shown to reviewers), but in grouped data there was no significant difference between the two groups. Thus, it was hard to conclude anything from this additional experiment.

Due to the lack of comprehensive PDX efficacy data, we decided to conduct additional fully human experimentation to replace the animal experiment but still add more patient relevance to our study. To achieve this, we incorporated two *EGFR*-mutant patient tumor-derived organoid (PDTO) lines into a novel tumor-on-a-chip (TOC) system developed within our laboratory (detailed method paper under review, TOC systems in general reviewed by for example Liu et al, (2)). This fully human system is designed to simulate several key aspects of *in vivo* tumor environments, including drug distribution dynamics and interactions within a three-dimensional extracellular matrix (3D-ECM). A critical feature of our model is the incorporation of an endothelial barrier within the chip, which the therapeutic agents must penetrate before reaching the tumor organoids. This setup not only adds a layer of complexity to the drug delivery process but also enhances the biological relevance of the model by mimicking the *in vivo* conditions more closely.

In the TOC, we observed that the *EGFR*-mutant PDTOs were sensitive to osimertinib, but they were even more responsive to the combination of osimertinib with panobinostat. The newly generated data from the TOC has been added to **Figure 7K**. We hope that this newly generated data is sufficient to show that the combination provides therapeutics benefits also in a patient-derived model.

Figure 7K

Reviewer #3:

Alsaed et al present a well-written manuscript concerning the heterogeneous expression level of mutant EGFR in lung cancer, and address the functional consequence of having a minor cell population with low mEGFR expression.

Overall, this is a good manuscript that is quite extensive from observation, to mechanism, to therapeutic intervention. However, on various key claims the authors remain rather correlative, while a more direct assessment is warranted. Below are a few major comments to improve the support for their claims.

1. It is of interest to understand the molecular mechanism why the mEGFR^{low} cells are resistant towards EGFR inhibition. In particular, how relevant is the EGFR-MAPK signaling pathway in tumor cells with low mEGFR expression (only assessment that is currently present is a EGFR gene signature)? The authors should characterize pathway components using biochemistry, including effects of inhibiting EGFR, pan-HER, pan-RAF, MEK or ERK? In parallel, perform dose-response measurements against pan-RAF, MEK and/or ERK inhibitors?

Response:

We appreciate this detailed comment from the reviewer. To address this, we have performed multiple new experiments to address further the signaling and dependencies of the mEGFR-low vs mEGFR-high cells. First, we assayed the cells with two protein expression arrays to compare signaling differences between parental, mEGFR-low, and mEGFR-high cells (in the picture below, pathways marked in red are the ones that seem to be upregulated in mEGFR-low and green ones downregulated). We observed a slight upregulation of AKT and p70S6K in mEGFR-low cells, indicating that they could be more reliant on AKT signaling. There was also additional interesting differentially expressed pathways in the low cells, including insulin and IGF1R signaling, but we feel this exploration is outside the scope of this particular study (data only shown to reviewers).

After these observations we conducted more focused western blot analysis related to AKT and MAPK signaling. Here the difference in AKT signaling was not as clear, but we did observe reduced MAPK (pMEK, pERK) signaling present in the mEGFR-low cells, suggesting they could be less reliant on MAPK compared to the mEGFR-high cells. When testing the mEGFR-low vs mEGFR-high cells for MEK and AKT inhibitors, as observed earlier with the IC50 testing (**Supplementary Figure 3B**), these slight differences were not enough to induce significant changes in the MEK/AKT inhibitor sensitivities (shown below). New piece of data about AKT/MAPK signaling was added to **Supplementary Figure 3D-F**.

2. I am very intrigued by the origin of the EGFR^{low} cells. Can the authors include lung cancers in their analyses that are wild-type for EGFR, as well as normal lung epithelium? Are low EGFR expressors to be detected in these cancer subtype or normal lung epithelium as well? If so, do they share similarities with the low mEGFR cells as described in the manuscript?

Response:

Thank you for this insightful comment! We have now approached this question from multiple directions. We first conducted immunohistochemical staining for EGFR in normal lung tissues. Even though EGFR staining was lowly present in the normal lung tissues, it was hard to further analyze expression heterogeneity or expressing cell types based on the staining. In general, expression was observed both in alveoli and bronchiole. To clarify these findings, we further analyzed single cell RNA sequencing data derived from normal lung. In the single cell seq, EGFR expression was most present in the AT1 and AT2 subtypes of the alveoli, but some expression was present also in for example ciliated bronchial cells of the lung. Most interestingly, heterogeneity in EGFR expression as well as differentially clustering subpopulations of AT1 and AT2 cells were observed. As AT1 and AT2 have both been considered as potential cell types for tumor initiation in lung adenocarcinomas (3), the observed heterogeneity in EGFR expression could already pre-date the formation of EGFR mutations.

We also stained tissues for total EGFR in several patient NSCLC tumors wild type for EGFR. Here, we also observed heterogeneous expression of EGFR, even though the expression levels in general were much lower than typically observed in mutEGFR / EGFR amplified tumors. These observations make it likely that inter-tissue expression differences in EGFR can be part of normal lung biology, but this phenotypic plasticity / variability is then hijacked during tumorigenesis to be able to overcome EGFR inhibitor treatment.

New data regarding the normal lung EGFR expression has been added to **Supplementary Figure 1C-E**.

3. In relation to 2. scRNA-seq on enriched populations of low versus high EGFR expressors from in vivo (PDX) tumors will provide superior in-depth insights into the phenotype of the cells as they appear in vivo, and the signaling pathways/ gene networks that are active in these cells (e.g. MAPK signaling and/or EMT).

Response:

We are very grateful for this insightful comment. To better address the EGFR-low/high phenotypes, we conducted single-cell RNA sequencing (scRNA-seq) on two EGFR-mutated NSCLC tumors. Tumor cells were analyzed using the Seurat toolkit, identifying the EGFR-low cell population as those exhibiting the lowest 20% of EGFR expression, and the EGFR-high cell population as those with the highest 20% in each sample. Gene Set Enrichment Analysis (GSEA) was performed to explore pathways involved in differential expression between these subsets. We observed that the EGFR-high state correlates with ERK signaling, while the EGFR-low state associated with enhanced EMT and TGFβ gene expression signatures; however, some results were not significant defined by the FDR q value. Interestingly, we also detected increased KRAS signatures in the EGFR-low populations, suggesting potential differential oncogenic dependencies compared to the EGFR-high cells. While these observations from scRNA-seq support our hypothesis in general, we plan to further explore these phenotypes in detail in a continuation work (incl. bigger patient N numbers), as we believe extending this investigation in regards to this manuscript is beyond the scope of the current work (data below shared with reviewers only).

4. *The general assessment of EGFR expression towards TKI sensitivity is not informative. The point made by the authors is the existence of a minor cell population with low mEGFR expression. Expression level in these few cells is easily masked by expression levels in the dominant cell populations when bulk analyses are performed. Measuring the fraction of low EGFR cells within various cell populations might be more informative.*

Response:

Thank you for your comment. We appreciate the opportunity to clarify our methodology, as it seems there may have been a misunderstanding regarding our experimental approach. Contrary to the concerns raised, most of our signaling experiments were not conducted using bulk analyses which might mask the contributions of minor cell populations. Instead, we used separated cell populations to ensure that our measurements accurately reflected the behavior of distinct cellular subsets. Additionally, the incorporation of single-cell RNA sequencing data, as discussed in response to Comment 3, allows for the precise detection and analysis of cells with low EGFR expression, providing robust support for our claims. This methodology addresses the potential oversight of minor populations and enhances the informativeness of our findings regarding TKI sensitivity.

5. *The part that centers around TGFbeta2 expression is interesting, but remains very much descriptive and correlative. Generation of TGFbeta2 knock-outs are required in order to pinpoint all those phenotypic observations to TGFbeta2. Same holds true for FN1.*

Response:

We appreciate this comment from the reviewer and fully agree with them. To address this, we have generated siRNA KDs and CRISPR KO:s from *ITGb3*, *TGFb2* and *FN1* (the same data was also requested by reviewer 2, comment 3).

We observed, that knock down of either of the three genes was able to reverse the migratory phenotype (shown in **Supplementary Figure 5 A-B**, with the strongest phenotype reversal observed in TGFb2 KD followed by ITGB3), and KD/KO of the genes was able to better sensitize the EGFR-low cells to osimertinib (as shown in the new **Figure 4N**). Increased EGFR expression was associated with KD of all three genes, but the phenotype reversal was again most strong with TGFb2 KD (shown in **Supplementary Figure 4H-I**).

Supplementary Figure 5A-B

Figure 4N

Supplementary Figure 4H-I

6. Using epigenetic modifiers to restore *mEGFR* expression is interesting, but the downside is that pan-HDAC inhibitors have wide effects on all gene-expression networks and are thus not specific towards restoration of *EGFR* expression alone. Moreover, it seemed as if *EGFR* levels were boosted in both cell population (*EGFR*^{low} and high). For the conclusion that the cells become sensitive towards *EGFR* inhibition due to its pharmacological upregulation, it is required to restore/enhance *mEGFR* expression in a gene specific manner, e.g. using *CRISPRa*, in *EGFR*^{low} cells and show that this leads to same sensitivity towards *Osimertinib* when compared to the *mEGFR* high expressors. Currently, it cannot be excluded that the cells become sensitive towards *EGFR* inhibition due to the crude intervention into the cells epigenetic status and (potentially) accompanying induction of stress.

Response:

We agree with the reviewer on the broad impact of HDAC inhibitors also on other things beyond *EGFR* expression. To address the specificity regarding increased *EGFR* expression and *EGFR* TKI efficacy, we generated a conditionally degradable form of mutant-*EGFR* (deg-TAG E476-A750Adel, degraded by the presence of dTAG13 compound, plasmid system by *Nabet et al, 2018, Nat Chem Biol, PMID: 29581585 (4)*). The system was introduced to the PC-9 *EGFR*^{low} cells to induce ectopic expression of mutant *EGFR* and to

degrade it upon will. We also included a non-functional version of the system in which a STOP codon was introduced in the middle of the construct, thus preventing the conditional degradation of EGFR by the system (only expressing the mutant EGFR). We observed, that upon the presence of ectopic mutEGFR, the mutEGFR-low cells became more sensitive to osimertinib, similar to the mutEGFR-high cells. When EGFR was conditionally degraded, the cells were able to tolerate osimertinib, similar to the parental cell line. Same rescue effect was not seen with the construct where the STOP codon was introduced. We have added this new piece of data into **Supplementary Figure 7I-J**.

Minor comments

There are always pro's and con's about specificity of TKIs and which one is the best. By my knowledge, afatinib is not specific for EGFR (line 146). To avoid such discussions, can the authors include a whole family of EGFR and pan-HER inhibitors, including gefitinib, lapatinib, dacomitinib etc.

Response:

We fully agree with the reviewer and have corrected the wording to the manuscript.

*“Next, we wanted to investigate how EGFR protein status influences the response to TKIs. In a short-term IC_{50} assay, EGFR-low cells were slightly less sensitive to EGFR-inhibiting drugs osimertinib and **pan-ErbB TKI afatinib**...”* In silico analysis was performed in the paper about sensitivity of EGFR-low vs EGFR-high cell lines to various TKIs, and can be found from Supplementary Figure 3G-H.

Typo in line 92

Response:

We have corrected this typo, thank you for pointing this out.

References:

- 1) Massagué J, Sheppard D. TGF- β signaling in health and disease. *Cell*. 2023 Sep 14;186(19):4007-4037. doi: 10.1016/j.cell.2023.07.036. PMID: 37714133.
- 2) Liu X, Fang J, Huang S, Wu X, Xie X, Wang J, Liu F, Zhang M, Peng Z, Hu N. Tumor-on-a-chip: from bioinspired design to biomedical application. *Microsyst Nanoeng*. 2021 Jun 21;7:50. doi: 10.1038/s41378-021-00277-8. PMID: 34567763.
- 3) Li Z, Zhuang X, Pan CH, Yan Y, Thummalapalli R, Hallin J, Torborg S, Singhal A, Chang JC, Manchado E, Dow LE, Yaeger R, Christensen JG, Lowe SW, Rudin CM, Joost S, Tammela T. Alveolar Differentiation Drives Resistance to KRAS Inhibition in Lung Adenocarcinoma. *Cancer Discov*. 2024 Feb 8;14(2):308-325. doi: 10.1158/2159-8290.CD-23-0289. PMID: 37931288.
- 4) Nabet B, Roberts JM, Buckley DL, Paulk J, Dastjerdi S, Yang A, Leggett AL, Erb MA, Lawlor MA, Souza A, Scott TG, Vittori S, Perry JA, Qi J, Winter GE, Wong KK, Gray NS, Bradner JE. The dTAG system for immediate and target-specific protein degradation. *Nat Chem Biol*. 2018 May;14(5):431-441. doi: 10.1038/s41589-018-0021-8. Epub 2018 Mar 26. PMID: 29581585.

REVIEWER COMMENTS

Reviewer #4:

The paper by Alsaed includes an impressive amount of data that all together support important advances on the comprehension of the role of EGFR-low cell in TKI resistance in lung cancer. Moreover, a potential pharmaceutical strategy is suggested to sensitize these EGFR-low cells to TKI by epigenetic upregulation of EGFR.

Comment 1:

Among others, the authors exploit also microfluidic approaches to study invasion, immune killing and drug responses. While conclusions appear overall correct, some methodological information are missing, in particular regarding the TOC experiments with PDOs (Fig. 7K). Even though another detailed method paper is under review, readers of this article should be able to understand the presented experiments.

Response:

Thank you for the valuable comment. We have now included additional details on how the TOC experiments with PDOs were conducted in the Methods section of our revised submission.

Comment 2:

How was the presence of the HUVEC endothelial barrier verified in the reported experiments?

Response:

*Thank you for this comment. We have verified the presence of the endothelial barrier in several microfluidic systems using multiple methods. In this specific set-up, dextran green fluorescent dye (here: 500 kDa, although other molecular weights have also been tested) was injected into the right distal channel of the microfluidic devices with and without an endothelial barrier (shown in figure **A** below). We then measured dextran dye permeability towards the ECM (middle channel) and the distal left channel over 30 minutes. The dextran permeability is limited in the presence of an endothelial barrier (shown in **B**). Additionally, we stained the microfluidic chip with Live-Green/Dead-Red staining and acquired confocal images to verify the formation of endothelial tubules (shown in **C**). Previously we have also stained the endothelial cells with specific markers, such as CD31 and F actin. We have also addressed*

chemotherapy sensitivity with and without the barrier, example data is shown in **D**. Small piece of data regarding the validation has been added to **Supplementary Figure 7F**.

Comment 3:

Do organoids expand in the TOC collagen over the 7 days?

Response:

Thank you for your comment. Tumor cells are seeded as 3-5 cell aggregates, allowing them to form organoids within three days. Post-formation, organoids in certain conditions, such as the control, can expand until day 7. This expansion is considered when evaluating drug efficacy, as the live area of organoids in each condition on day 7 is normalized to the live area in the control condition.

Comment 4:

At what concentration (mg/ml) does 80% collagen correspond? Which green dye is used to stain live organoids?

Response:

The 80% collagen corresponds to 4 mg/ml in PBS. This detail has been added to the corresponding Methods section in the revised submission.

Organoids were stained with Live-Green/Dead-Red AO/PI staining. This detail has now been added to the corresponding Methods section in the revised submission.

Comment 5:

Please for clarity remove mention to drug distribution dynamics and interaction with 3D-ECM (lines 409-410), since it was not assessed in this work.

Response:

The statement has been revised in the new submission.

Comment 6:

Please, improve the paragraph 'Drug testing in tumor-on-a-chip' (lines 545-556). It is written that middle channel was seeded with tumor cells, but it should be with pre-formed organoids.

Response:

Thank you for the valuable comment, the paragraph has been improved, please see our responses for comments 1 and 3.

Comment 7:

Fig. 5E, please define what is '%relative invasion' on y axis.

Response:

Thank you for the comment, it's now defined in the figure legend of the new submission.

Comment 8:

One of the paper conclusion is that EGFR-low cells exhibited an EMT phenotype. It would be informative to see pictures of EGFR-low cells showing a more mesenchymal phenotype than EGFR-high cells. Does the RNA sequencing of low versus high show upregulation of the typical EMT transcription factor signatures (Slug, Snail, Twist, Zeb)?

Response: We thank the reviewer for this kind and interesting comment! Like shown in the paper, we constantly detect hallmarks of EMT in the EGFR-low cells, including lower E-cadherin expression and increased SMA in the protein level, as well as increased FN1 and TGFb secretion also shown in the manuscript. This to us suggested at minimum, “EMT-like” phenotype typical for invasive cancer cells (I.e. Nieto et al, Cell, 2016 PMID: 27368099, additional western blot run was performed below).

In the mRNA sequencing data, we additionally found that EGFR-low cells express more EMT markers Vimentin (VIM), Snail (SNAI1) and Fibronectin1 (FN1). Slug (SNAI2), however was more expressed in the EGFR-high cells, possibly agreeing with our hypothesis that the EGFR-low cells are not “full blown” EMT phenotype, but somewhere in between. ZEB1 and ZEB2 (as well as the TWIST genes) were not expressed in the cells, also agreeing with this theory. The term “EMT-like phenotype” instead of “EMT phenotype” is also used in the manuscript for the same reason. We also have added a small piece of text to the manuscript stating the partial modulation suggested by the new E-Cadherin data (below). We also found this phenotypic plasticity highly interesting, and would like to pursue it further in the follow up investigations.

Comment 9: The model of Fig. 7L (whose legend is missing) reports upregulation of E-cadh upon HDACi, but WB of Fig. 6B does not include E-cadh.

Response:

Thank you for this observation. We deeply apologize that Figure. 7L was missing the legend, it's now added to the new submission. We have also now addressed the E-cadh expression after HDACi treatment and observed no notable changes (at least upon 1-5 day treatment period). Our interpretation here is that HDACi is able to induce some phenotypic changes (like upregulation of EGFR and downregulation of SMA, shown in Fig 6B), but it is not able to fully alter the EMT-like phenotype. New data and text has been added to **Supplementary Figure 6D** and the **Figure 7L** has been changed according to our new findings.